# MYC2: A Master Switch for Plant Physiological Processes and Specialized Metabolite Synthesis

**DOI:** 10.3390/ijms24043511

**Published:** 2023-02-09

**Authors:** Lei Luo, Ying Wang, Lu Qiu, Xingpei Han, Yaqian Zhu, Lulu Liu, Mingwu Man, Fuguang Li, Maozhi Ren, Yadi Xing

**Affiliations:** 1Zhengzhou Research Base, State Key Laboratory of Cotton Biology, School of Agricultural Sciences, Zhengzhou University, Zhengzhou 450001, China; 2State Key Laboratory of Cotton Biology, Institute of Cotton Research, Chinese Academy of Agricultural Sciences, Anyang 455000, China; 3Institute of Urban Agriculture, Chinese Academy of Agricultural Sciences, Chengdu 610000, China; 4Hainan Yazhou Bay Seed Laboratory, Sanya 572000, China; 5School of Pharmaceutical Sciences (Shenzhen), Shenzhen Campus of Sun Yat-sen University, Shenzhen 518107, China

**Keywords:** MYC2, jasmonic acid signaling, plant physiology, specialized metabolites, biosynthesis, synthetic biology

## Abstract

The jasmonic acid (JA) signaling pathway plays important roles in plant defenses, development, and the synthesis of specialized metabolites synthesis. Transcription factor MYC2 is a major regulator of the JA signaling pathway and is involved in the regulation of plant physiological processes and specialized metabolite synthesis. Based on our understanding of the mechanism underlying the regulation of specialized metabolite synthesis in plants by the transcription factor MYC2, the use of synthetic biology approaches to design MYC2-driven chassis cells for the synthesis of specialized metabolites with high medicinal value, such as paclitaxel, vincristine, and artemisinin, seems to be a promising strategy. In this review, the regulatory role of MYC2 in JA signal transduction of plants to biotic and abiotic stresses, plant growth, development and specialized metabolite synthesis is described in detail, which will provide valuable reference for the use of MYC2 molecular switches to regulate plant specialized metabolite biosynthesis.

## 1. Introduction

Phytohormones are important regulators of plant growth and development. In response to damage, plants generally integrate phytohormone signaling pathways to trigger immune defense and repair responses throughout the plant body [1]. Particularly, jasmonic acid (JA) signaling is a core signaling pathway that becomes activated in response to plant damage [2]. After the plant is damaged, a large amount of JA is produced, which is then transformed into (+)-7-iso-jasmonoyl-L-isoleucine (JA-Ile) with biological activity. JA-Ile promotes the formation of the SCF-COI1-JAZ coreceptor complex, degrades JAZ (Jasmonate ZIM-domain) through 26S proteasome ubiquitination, and relieves JAZ’s inhibition of MYC2, thereby activating the transcription of MYC2 downstream genes, thus triggering plant defense and repair [3]. As a JA signaling hub, MYC2 participates in multiple signaling pathway-networks that integrate light signaling, hormone signaling, natural product synthesis, and the complex processes of plant growth and development [4,5,6,7,8,9]. In addition, JA not only activates the defense response of the plant itself, but also enhances the content of linalool [10] and β-ocimene or induces the release of other volatile organic compounds, such as shikimic acid derivatives, to enhance the resistance of adjacent plants to the attack of arthropod herbivores [11].

In contrast to animals, plants are sessile and cannot flee from biotic or abiotic stress conditions [12]. Plants therefore produce specialized metabolites to respond to stress; indeed, when subjected to stress, plants generate and accumulate specialized metabolites to improve their immunity and resistance [13]. Plant specialized metabolites not only induce plant resistance to stress but are also widely used in the chemical, food, and agriculture industries, especially in the medical field. They can be divided into tannins, flavonoids, terpenoids, alkaloids, quinones, etc. At present, the most important metabolites from plants are taxol, vinblastine, camptothecin, and artemisinin [14,15]. For example, paclitaxel and vinblastine are effective in the treatment of cancer, while artemisinin is widely used in the treatment of malaria [16]. Secondary metabolites of plants can be extracted by distillation, organic extraction, solid phase extraction, super critical fluid extraction, pressurized liquid extraction, and microwave-assisted extraction, but they are restricted by the low yield and long plant growth cycle [17]. Chemical synthesis of natural products not only faces challenges related to the uncertain structures and complex chiral centers of natural products, but additionally involves cumbersome steps, shows a low conversion rate, and consumes high amounts of energy. Therefore, it is difficult to meet market demands using direct extraction or chemical synthesis.

Synthetic molecular switches are mostly based on protein–protein, protein–DNA, or protein–RNA interactions [18]. Plant transcription factors, as natural regulatory molecular switches, are controlled by endogenous small-molecule metabolites, exogenous damage patterns, and external stimuli, such as light, to regulate downstream genes [19]. Thus, we can use synthetic biology to separate and represent plant functions. The transcription factor MYC2 is a high-level transcriptional regulatory element in the JA signal pathway. MYC2 and its direct binding secondary transcription factors form a series of transcriptional cascade regulatory modules, which activate and amplify the response caused by JA and can achieve the effect of regulating multiple transcription factors [20]. Harnessing the phenomenon transcription factor MYC2 can be precisely regulated by JA signals; the JA-JAZ-MYC2 regulatory pathway was developed into a controllable molecular switch and applied in synthetic biology to precisely regulate the biosynthesis of plant secondary metabolites. The synthetic gene of the target natural product was used to construct an artificial chromosome, and the JA-JAZ-MYC2 switch was added to transform chassis cells or plants to achieve efficient synthesis of the target product. In this review, we systematically analyze and summarize the molecular mechanisms of plant MYC2-regulation of physiological and biochemical processes and provide a reference for the assembly and construction of new plant metabolite production models by characterizing plant signaling pathways.

## 2. MYC2-Mediated JA Signaling

Jasmonic acid is the first hormone to respond to and rapidly accumulate after wounding. The core JA signaling module includes the F-box protein SCF^COI1^, JAZ, and the transcription factor MYC2 [21]. In the JA signaling pathway, MYC2 is the main downstream effector, and JAZ is a negative regulator of JA signaling that binds to MYC2, thereby inhibiting its transcription factor activity. MYC2 is a member of the basic helix loop helix (bHLH) family of proteins, which consist of a basic domain and a bHLH domain. The N-terminus of MYC2 is composed of a JAZ interaction domain and a transcriptional activation domain, and the C-terminus is a DNA-binding bHLH domain that can bind to DNA and combine with the *cis*-acting G-box element (5′-CACGTG-3′) [22]. In *Arabidopsis*, the enzymes acyl-CoA oxidase, nucleoside hydrolase 3, oxophytodienoate reductase 3 (OPR3), alkene oxide synthase (AOS), and dihydroflavonol 4-reductase all contain G-box elements in their promoter regions and are induced by JA (https://www.arabidopsis.org/ (10 November 2022)). In the absence of JA, JAZ interacts with MYC2 and recruits the corepressor Topless through NINJA (novel interactor of JAZ), inhibiting the expression of JA-responsive genes activated by MYC2 [23]. Consistent with this, a single amino acid change in MYC is sufficient to cause the loss of interaction with most JAZ proteins (except for JAZ1 and JAZ10). For example, when D105 of MYC2 or D94 of MYC3 mutates, its interactions with most JAZ proteins are eliminated, thus affecting the trans activation activity of MYC [24]. Transcriptional repression by JAZ also involves the recruitment of chromatin-modifying proteins. Furthermore, JAZ interacts with histone deacetylase 6, leading to a closed chromatin state and repression of the transcription of JA-responsive genes [25]. When JA accumulates, it forms a biologically active JA-Ile, which is perceived by COI1 and forms a platform to recruit JAZ, and then degrades it through ubiquitination modification, releasing transcription factors and initiating JA-regulated plant reactions [26]. MYC2 forms homo- or heterodimers with MYC3 and MYC4, and MYC5 binds to the conserved G-box present in the promoters of JA-responsive genes [27]. MYC2 can activate downstream response genes by modifying histones. For example, it can affect histone H3 methylation of the salt stress response gene [28]. MYC2 can also form a complex with the MED25 subunit of the medium complex, thus recruiting the histone acetylase HAC1, histone acetyltransferase GCN5, and nucleosome remodeling protein SPLAYED to the promoter of downstream genes and thereby selectively regulating the state of target histones [29,30]. By interacting with the MED25 subunit of the mediator complex, MYC2 also recruits the pre-transcription initiation complex, the mediator complex, RNA polymerase II, and other components of the general transcription factor to target promoters [31]. Furthermore, MYC2 can activate JA-induced bHLH proteins (MTB1–3) [32,33] and regulate JA signaling via a negative feedback by impairing MYC2-MED25 complex formation, leading to the termination of JA signaling [34]. LUH activates MYC2-directed transcription of *JAZ2* and *LOX2* via the mediator complex coactivator and the histone acetyltransferase HAC1. We showed that the mediator subunit MED25 physically recruits LUH to MYC2-target promoters, thereby linking MYC2 to HAC1-dependent acetylation of Lys-9 of histone H3 (H3K9ac) to activate *JAZ2* and *LOX2*. LUH interacts with MED25 and HAC1 through its distinct domains, thus imposing a selective advantage by acting as a scaffold for MYC2 activation [30].

The crosstalk/interaction between MYC2 and abscisic acid (ABA), ethylene (ET), GA, and SA mediate various plant developmental processes and defense responses. The signal transduction of ABA and JA is connected through the direct interaction between the ABA receptor PYL6 (RCAR9) and MYC2 [35]. In the ET signaling pathway, the interaction between MYC2 and EIN3 regulates the antagonism of JA and ET signals. MYC2 inhibits the transcriptional activity of EIN3 and EIL1 according to pull-down experiments. It also inhibited the expression of wound response genes and herbivore-induced genes induced by JA by interacting with EIN3 and inhibiting its DNA binding activity, and alleviated the defense of plants regulated by JA against herbivores [36]. The DELLA protein is a key inhibitor of the GA signal pathway, and it can directly interact with JAZ1, prevent JAZ1 from interacting with MYC2, and enhance the binding of MYC2 to its target gene promoter to promote JA reaction [37]. The SA and JA signaling pathways are known to intersect at various points because the two regulate biotic stress responses antagonistically [38]. Studies have shown NPR1 (which can activate SA response genes) to be a key player in the antagonistic crosstalk between SA and JA. The SA-facilitated suppression of JA-responsive genes, such as *LIPOXYGENASE 2* (*LOX2*), *VEGETATIVE STORAGE PROTEIN* (*VSP*), and *PDF1.2*, was abolished in npr1 mutant plants [39]. MYC2 can participate in the regulation of the crosstalk between the SA and JA pathways. In the presence of SA, NPR1 can interact with JA-induced MYC2 and inhibit the transcriptional activation of downstream response genes by interrupting the interaction between MYC2 and MED25 [40].

In addition to the regulation of MYC2 by JAZ, photoreceptors could be involved in blue-light sensing and the MYC2 pathway, and this phenomenon is coordinated with photoperiod or circadian responses. For example, a blue light signal can induce the expression of MYC2/MYC4 through CRYPTOCHROME1 (CRY1) signal transduction to activate NST1, further activate the transcription network of the secondary cell wall (SCW), and regulate the thickening of SCW in fibroblasts [41]. MYC2/ZBF1 can also mediate the interaction between light signals and JA signals, participate in the regulation of the cryptochrome-mediated blue light signal pathway, and play a role in blue light-mediated photomorphological growth. When plant seedlings are exposed to light, the blue light receptor CRYs sense the light signal and induce the biosynthesis of JA, thus activating the transcription factor MYC2/3/4, further activating the expression of the light morphogenesis regulatory gene *HY5*. In addition, *cry1* and *cry2* photoreceptors can activate the negative blue light response regulator MYC2, which can directly blind to the G-box in the plant pigment SPA1 promoter to regulate its expression [42]. SPA1 is a protein involved in regulating circadian rhythm and light signal transmission. MYC2 and SPA1 can inhibit light morphological growth, negatively regulate blue light-mediated light morphological growth, and inhibit blue light-regulated gene expression in the dark [43]. The circadian response enables plants to best respond to environmental challenges. The biosynthesis of JA is controlled by the biological clock, and its accumulation follows the law of rising in the day and falling at night. Therefore, many transcription factors, including MYC2, regulating JA signaling are also controlled by the biological clock and are governed by similar laws [44]. Inhibition of MYC activity is therefore necessary to reset JA signaling and avoid deleterious runaway responses. The involvement of SUMOylation in the modulation of MYC2 activity has recently been reported. Researchers found that blue light exposure enhances the SUMOylation of MYC2 in K139 and K480. SUMO conjugation does not affect the dimerization of MYC but modulates the interaction of MYC2 with its cognate DNA cis-element and with the ubiquitin ligase plant U-box 10 (PUB10). Moreover, the non-SUMOylatable MYC2 is less stable and interacts more strongly with PUB10; thus, SUMO functions as a counterpoint to the ubiquitin-mediated degradation of MYC2 [45]. In addition, E3 ubiquitin ligase can also reduce MYC protein level based on Cullin3 and BPM proteins as substrate junctions. JA enhances BPM3 stability and establishes a negative feedback regulatory loop to control MYC level and activity. In this new JA pathway regulatory layer, MYC is degraded and terminated by CUL3 BPM-mediated MYC activity [46].

## 3. MYC Is Involved in the Regulation of Biotic and Abiotic Stress Conditions

Plants have developed numerous strategies to cope with biotic and abiotic stresses. Trichome is the first defense layer of plants against biotic and abiotic stresses. AtMYC1, which positively regulates trichome initiation by regulating the intracellular localization of GL1 and TRY, has been shown to participate in the regulation of trichome patterning [47,48,49]. Plants can also regulate immune responses against different pathogens by releasing specialized metabolites [50,51]. Phytohormone signaling pathways mediated by JA play a key role in many of these defense responses. When attacked by herbivores, plants activate a series of synthetases to trigger JA biosynthesis, including 13-lipoxygenase (13-LOX), AOS, oxyalkylene, cyclase, and OPR3 [52,53,54]. This in turn triggers a series of JA-dependent downstream reactions, such as the biosynthesis of several metabolites and molecules, such as the defense protein thionine (encoded by THI) and nutrient storage protein 1 (encoded by VSP1) [55,56]. Jasmonates are major regulators of defense responses, and a network centered on MYC2 is involved in the defense response to multiple pathogens [57]. MYC regulates the expression of genes involved in the synthesis of various specialized metabolites related to insect resistance [58]. For example, in *Arabidopsis*, AtPIFs and AtMYC2 form a homodimer that binds to the promoter of the *TPS* terpene synthase gene to increase terpene biosynthesis as a defense (Figure 1a) against whitefly (*Bemisia tabaci*) [59]. Similarly, MYC-related mutants reportedly show differential resistance to cotton bollworm (*Helicoverpa armigera*), which may be related to the roles of *MGAIs* (MYC-related genes against insects) in flavonoid biosynthesis [58]. High concentrations of flavonoids inhibit the growth of certain insects and allow plants to recover after injury or insect attack. MYC2 may regulate flavonoid production by acting on MGAIs to alleviate plant growth inhibition caused by insect damage [58]. MYC1 is involved in the regulation of the flavonoid biosynthesis pathway and epidermal cell fate in grapevine [60]. By analyzing resistance indicators of *Solanum lycopersicum* strains inoculated with *Meloidogyne incognita*, MYC2 was found to negatively regulate the sensitivity of *Solanum lycopersicum* to *M. incognita* [61]. Nematode RALF-like peptide binds to the extracellular receptor domain of FER, triggering MAPK phosphorylation, JA signaling, and a reactive oxygen species burst, promoting the degradation of MYC2 to suppress plant immunity [62]. JA is a major regulator of *Nilaparvata lugens* resistance in rice, and rice *myc2* mutants are more susceptible to planthoppers than wild-type (WT) plants [63]. Members of the Solanaceae family accumulate phenylpropanoid-polyamine conjugates (PPCs) in response to attackers while also maintaining a chemical barrier of steroidal glycoalkaloids (SGAs). *Solanum lycopersicum MYC1* and *MYC2* redundantly control jasmonate-inducible PPC and SGA production and are also essential for constitutive SGA biosynthesis [64,65]. In addition, MYC is involved in regulating the expression of disease resistance related genes to regulate plant resistance to diseases. In *Solanum lycopersicum*, magnesium oxide positively regulates resistance genes by triggering JA signaling and activating MYC2 to induce immunity to *Fusarium wilt* [66]. Overexpression of *OsMYC2* induces up-regulation of *PR* gene and bacterial blight resistance in rice [67]. At the same time, MYC2, as a negative regulator of plant pathogens, can block the function of EIN3 by reducing the ET-mediated pathogen reaction. Further, MYC2 can interact with EIN3 to antagonize and regulate the expression of *ERF*, thereby enhancing the expression of *PDF1.2* induced by pathogens [68].

MYC plays important roles not only in plant immunity to pests and pathogens [69] but also in abiotic stress tolerance [70]. Wind and rain cause short-term molecular changes and have long-term developmental effects on flowering time, pathogen defense, and plant structure. Water-spray stress to simulate rainfall induced the activation of JA signaling, in which MYC2 activated ORA47 by interacting with bHLH19 and ERF109 to further promote JA synthesis, suggesting that water spray-induced JA accumulation is regulated by the MYC2-ORA47 pathway via positive feedback regulation [71]. MYC2 acts as a transcriptional activator in the ABA signaling pathway under drought stress [72] and positively regulates the ABA-inducible genes *rd22* and *ADH* [73]. Bayesian network models have suggested that MYC2 and ATAF1 may be regulators of drought response, and activation of MYC2 or inhibition of ATAF1 is the best single-node intervention strategy to modulate drought response [74]. MYC2-like can mediate the expression of *OsCYP2*, a cyclophilin chaperone. Further, *OsCYP2* has been proven to promote the resistance to abiotic stress when overexpressed, especially salt stress. In addition, MYC2-like is also a potential regulator in rice to regulate physiological processes related to salt stress through the abscisic acid (ABA) signaling pathway. A MYC2-like transcription factor binding to ABRE was also identified by yeast one hybrid assay and EMSA. An overexpression transformant of this transcription factor showed higher antioxidant enzyme activity in reactive oxygen species clearance. It is speculated that MYC2-like can improve the resistance to salt stress by improving the antioxidant enzyme activity and post-translational regulation of transformed plants [75]. The ICE1-CBF transcriptional cascade plays central roles in cold tolerance and cold acclimation in plants, and JAZ interacts with ICE to block the ICE-CBF pathway [76,77]. In apple, MdMYC2 interacts with the G-box in the *MdCBF1* promoter to regulate freezing tolerance (Figure 1a) [78]. Plants respond to harsh circumstances by producing specialized metabolites. For example, in cold-exposed sugar beet, betaine aldehyde dehydrogenase (PtrBADH-1) is activated by PtrMYC2 to regulate cold-induced glycine betaine accumulation [79]. At the same time, JA can also induce *SN13* expression in rice, thereby improving rice resistance to drought, cold, and freezing by affecting rice membrane integrity and osmotic matter accumulation [80].

## 4. MYC2 Is Involved in the Regulation of Plant Growth and Development

Through chloroplast photosynthesis and transpiration, leaves provide energy for plant growth, development, and reproduction. Leaf senescence is regulated by numerous signals, with complex crosstalk between different signaling pathways. MYC2 can inhibit leaf vein development by negatively regulating the biosynthesis of tryptophan, a key substrate for auxin biosynthesis (Figure 1b) [81]. Compared with WT plants, *Arabidopsis myc2* mutants have higher auxin contents and a significantly increased leaf vein density [81]. Further, MYC2 promotes the expression of the JA-induced senescence-related gene *Dof2.1*, and *Dof2.1* enhances JA-induced leaf senescence by directly activating the *MYC2* promoter (Figure 1b) to form a MYC2-Dof2.1-MYC2 feedback loop [82]. In senescent leaves, JA induces H_2_O_2_ accumulation via binding of MYC2 to the *CAT2* promoter to inhibit its expression, thus upregulating the expression of the senescence-related genes *SAG12*, *SAG13*, *SAG29*, and *SAG113* and inhibiting (Figure 1b) that of the photosynthesis-related genes *CAB1* and *RBCS* [83,84]. In turn, MYC5 functions redundantly with MYC2, MYC3, and MYC4 to modulate the expression of JA-regulated senescence- and photosynthesis-related genes, thus triggering JA-induced leaf senescence. In *Arabidopsis*, MeJA treatment activated MYC2 to promote the transcription of *AtUSR1* (Figure 1b), which is involved in age-dependent and dark-induced leaf senescence [85]. During the senescence of rape leaves, MYC can be induced and activated by ABA, and then up-regulate the expression of AMY3 and BAM1, which are related to starch degradation, and the sucrose transporters SUT1, SUT4, and SWEET11 [86]. In the process of *Solanum lycopersicum* leaf senescence, SlMYC2 can enhance the expression of *SIPAO*, which encodes a chlorophyll degrading enzyme and plays an active role in *Solanum lycopersicum* leaf senescence [87]. These findings suggest that MYC2 plays an important regulatory role in plant leaf senescence, and thus identification and characterization of key senescence genes regulated by MYC2 are of great significance to understand the molecular mechanisms underlying leaf senescence.

The plant root system consists of taproots, lateral roots, and adventitious roots, which play important roles in plant anchoring, water and nutrient absorption, and the synthesis of growth regulators [88]. In *Arabidopsis*, JA inhibits cell division in the main root meristem in a MYC2-dependent manner. MYC2 directly binds to the promoters of *PLT1* and *PLT2* to inhibit their expression and thus negatively regulates root stem cell maintenance and root meristem activity [88]. The transcription factor ethylene response factor 115 (ERF115) is a rate-limiting factor of quiescent center cell division, which is controlled via transcriptional activation of phytosulfonamide signaling. JA induces ERF115 expression (Figure 1d) in a COI1- and MYC2-dependent manner [89]. In root regeneration, JA signaling induces cell proliferation by activating MYC2 and restores root meristems via ectopic induction of ERF115. Ethylene-responsive factor 109 (ERF109) directly binds to the GCC-box in the *ASA1* promoter and is involved in regulating lateral root formation (Figure 1d) in *Arabidopsis* [90], in which JA activates MYC2 to upregulate auxin synthesis by activating ERF109 and promotes de novo root regeneration in response to injury [71]. Plant roots have evolved different systems for nitrogen uptake. Auxin, CTK, ABA, ETH, GA, BR, and JA can regulate NO3− and NH4+ uptake by regulating the transcript levels or transport activities of the NPF/NRT1, NRT2, and AMTs families in various plants [91]. In addition, JA can also mediate the expression of OsWRKY28 to enhance rice root elongation and phosphate absorption [92]. Simultaneously, JA can also enhance the nitrogen absorption of legumes by enhancing the formation of root nodules, thus promoting the growth of roots [93].

In apple, JA and MYC2 promote the expression of 1-aminocyclopropane-1-carboxylic acid oxidase and synthase and ET synthesis (Figure 1c) to promote fruit ripening [94]. MYC2 can not only directly promote the expression of 1-aminocyclopropane-1-carboxylic acid oxidase and synthase, but also promote the expression of 1-aminocyclopropane-1-carboxylic acid synthase via the ERF3 pathway. MYC2 can interact with ERF2 to relieve the binding between ERF2 and ERF3 [95] and enhance the transcriptional activation of *ACS1* by ERF3 [94]. In loquat fruit treated with recombinant serine protease, *MYC2* expression was upregulated, and as a result, the expression of genes encoding polyphenol oxidase, phenylalanine ammonia lyase, and other resistance-related genes was upregulated, contributing to the resistance of fruit to external biotic stress and effectively delaying fruit rotting [96]. ABA induces the expression of *MYC2* and *MYB1R1* and activates the *PbFAD3a* promoter, contributing to the formation of russet pear skin [97]. MeJA treatment and SlMYC2 overexpression inhibited *Solanum lycopersicum* seedling growth and photosynthesis, but increased the sugar acid ratio and content of lycopene, carotenoid, soluble sugar, total phenols, and flavones, indicating that JA signal transduction could inhibit *Solanum lycopersicum* seedling growth and change fruit quality [9]. The process of ovule abortion is highly coordinated and controlled by numerous environmental and endogenous signals. JA and ET participate in the gibberellin-induced ovule programmed cell death process in seedless pear ‘1913’ (pyrus hybrid) [98]. The binding of MYC2 to the *SENESCENCE-ASSOCIATED 39* (*SAG39*) promoter triggers its expression to regulate ovule abortion [98]. In *Arabidopsis*, MYC2 binds to the *SAG29* promoter to activate its expression and thus trigger JA-induced leaf senescence [84,96]. Furthermore, *SAG39*, a homolog of *SAG29*, is expressed in ovules, suggesting that there may be similar regulatory mechanisms underlying leaf and ovule senescence. These results provide a strong basis for understanding ovule development for the breeding of seedless fruit.

Seeds are unique organs that consist of a seed coat, an embryo, and an endosperm. The embryo stores most of the nutrients required for plant germination, such as protein, starch, and fat stored in the form of triacylglycerol. In *Arabidopsis*, triacylglycerol and related storage proteins are rapidly synthesized and accumulated in early embryonic seed development, and the genes encoding seed storage proteins 12S globulin and 2S albumin may be regulated by MYC2 [99]. MYC2, MYC3, and MYC4 play synergistic roles in determining seed size, weight, and storage protein content. The weight and storage protein content of *Arabidopsis* triple mutant *myc2myc3myc4* seeds were significantly higher than those of WT seeds. In addition, CRA1 and CRU3 contents and their corresponding gene expression levels were also substantially higher than those in WT seeds [100,101]. The spikelet is the basal unit of inflorescence in grasses, and its formation is crucial for reproductive success and cereal yield. A study found that JA plays a key role in determining rice (*Oryza sativa*) spikelet morphogenesis. *Extra glume 1* (*eg1*) and *eg2* mutants exhibit altered spikelet morphology, with changed floral organ identity and number, as well as defective floral meristem determinacy. EG1 is a plastid-targeted lipase that participates in JA biosynthesis, and EG2/OsJAZ1 is a JA signaling repressor that interacts with a putative JA receptor, OsCOI1b, to trigger OsJAZ1 degradation during spikelet development. OsJAZ1 also interacts with OsMYC2 and represses OsMYC2’s role in activating OsMADS1, an E-class gene crucial for spikelet development [102]. NtMYC2a plays an important role in carbohydrate metabolism and pollen development by regulating the expression of the starch metabolism-related genes *AGPs*, *SS2*, and *BAM1* in pollen grains, anther walls, and ovaries of tobacco plants. The process of pollen maturation was accelerated in NtMYC2a-OE plants and was delayed in NtMYC2a-RI plants [103]. Chestnut (*Castanea mollisima*) is an important woody food crop, but its yield is low when cultivated, mainly due to the problems of fewer female flowers and more male flowers. A higher concentration of JA-Ile is conducive for the differentiation and formation of female flower buds during post-winter, and JAZ1-3 and MYC2-1 play a key role in the differentiation of female flower buds of chestnut [104]. Additionally, MYC2, MYC3, MYC4, and MYC5 have redundant roles in regulating stamen development and seed production, and can interact with MYB21 and MYB24 to form a bHLH-MYB transcription complex that regulates stamen development and seed production [105,106]. While inhibition of the bHLH-MYB complex by JAZ proteins inhibits stamen development and seed production, JA induces JAZ degradation and releases the bHLH-MYB complex, thus activating the expression of downstream genes critical for stamen development and seed production [107].

## 5. MYC2 Is Involved in the Regulation of Specialized Metabolites in Plants

JA can induce the biosynthesis of medicinally active components such as paclitaxel, artemisinin, tanshinone, and vinblastine in plants [53]. As the main regulator of JA signaling, MYC2 is involved in regulating the expression of key gene-encoding enzymes involved in the production of various plant specialized metabolites [108,109] and directly or indirectly affects the synthesis and accumulation of specialized metabolites [110]. In *Taxus chinensis*, TcMYC2 directly activates the expression of paclitaxel biosynthesis genes or regulates paclitaxel biosynthesis through ERF regulators (Figure 2a) [111]. When TcMYC2a was overexpressed in yew cells, the expression levels of paclitaxel biosynthesis-related genes significantly increased. TcMYC2 activates paclitaxel biosynthesis in response to MeJA and binds to the promoter of the paclitaxel biosynthesis gene encoding taxadiene synthase. Overexpression of *TcMYC2a* increases the expression of *TcERF12* and *TcERF15* to regulate taxadiene synthase gene expression [111]. In *Artemisia annua*, AaMYC2 directly regulates G-box-like elements in the promoters of *CYP71AV1* and *DBR2*, encoding two key structural enzymes in the artemisinin synthesis pathway, to induce their expression (Figure 2b) [112,113]. The transcription factors *AaMYC2*, *AaNAC1*, and *AaHD1* have been overexpressed in *A. annua* to improve artemisinin biosynthesis. This not only increased the artemisinin content but also allowed plants to grow in abandoned saline-alkali soils and conferred a certain degree of herbicide resistance to them [114]. In *Salvia miltiorrhiza*, SmMYC2 reportedly upregulates the transcription of genes such as *SmGGPPS* to promote tanshinone synthesis and regulates the expression of *CYP98A14* by binding to its E-box to promote tanshinone synthesis (Figure 2c) [115]. In *Catharanthus roseus*, CrMYC2 affects the expression of *SLS*, *TDC*, and *LAMT* by regulating the transcription of *ORCA2*, *ORCA3*, and *ORCA4* to activate key steps in the terpenoid-indole alkaloid pathway, thereby regulating vinca alkaloid synthesis (Figure 2d), resulting in vinblastine accumulation [116]. Gossypol is a sesquiterpene lactone and an important antimicrobial specialized metabolite involved in cotton (*Gossypium hirsutum*) resistance to pathogen attack and insect damage. Gossypol is used as an anticancer drug and a male contraceptive [117]. GhMYC2 directly regulates the expression of key enzymes in the gossypol synthesis pathway, such as CYP71BE79, and participates in the regulation of gossypol synthesis. GhMYC2 silencing and overexpression negatively and positively affected the gossypol contents in cotton tissues, respectively (Figure 2e) [118].

In *Tripterygium wilfordii*, TwMYC2a and TwMYC2b negatively regulate triptolide biosynthesis by inhibiting TwTPS27a and TwTPS27b in hairy roots [119]. MYC2 directly regulates the synthesis of triterpenoid saponins of *Psammosilene tunicoides* by regulating the backbone-building enzymes β-amyrin synthase and squalene epoxidase in the triterpenoid saponin synthesis pathway [120]. In addition to regulating the synthesis of various medicinal natural products, MYC2 plays an important role in the quality improvement of various natural products with industrial application value [121]. For example, HbMYC2b significantly increased the content of small rubber granule-protein in rubber tree (*Hevea brasiliensis*) latex. The specific JA signal transduction module COI1-JAZ3-MYC2 exists in rubber tree milk-duct cells, and MYC2 enhances secondary milk duct differentiation and rubber biosynthesis by upregulating the expression of farnesyl pyrophosphate synthase and small rubber granule-protein genes [122].

## 6. Prospects of Application of MYC2 in Chassis-Based Synthesis of Natural Products

Synthetic biology aims to redesign existing natural biological systems to achieve target functions [123]. The concept is fully applicable to the engineering of plants to synthesize plant natural products. Plant synthetic biology follows the design-build-test-learn cycle of synthetic biology [124]. Chassis cells are the hardware foundation of synthetic biology [125]. However, the construction of chassis cells for plant natural-product synthesis remains challenging [126]. The development of life sciences has made available complete genome sequences, has provided valuable bioinformatics and genetic resources, and has deepened our understanding of the biosynthesis of natural products and the regulatory mechanisms at play [127,128,129,130]. Transcription factors exist widely in organisms and have natural advantages as molecular switches. They can change their own structures and regulate downstream genes by sensing changes in the concentration of target natural products in cells, thus converting product concentrations into specific signal outputs. The phenomenon that the transcription factor MYC2 can be precisely regulated by JA signaling suggests that the JA-JAZ-MYC2 regulatory pathway has the potential to be used as a molecular switch in synthetic biology to precisely regulate the biosynthesis of plant specialized metabolites. JA exists in seed plants, and JA and JA-Ile are also detected in *bryophytes* [131]. JA was even detected in *Escherichia Coli* and *Saccharomyces Cerevisiae* cultures [132]. The JA-JAZ-MYC2 system activates the corresponding biosynthetic system by applying high doses of JA. Under non-stress conditions, the impact of the low content of endogenous JA produced in the host organism on the JA-JAZ-MYC2 system may be inevitable but also limited. The widespread existence of JA suggests that it may be feasible to use the JA-JAZ-MYC2 system to regulate biosynthesis in eukaryotic and prokaryotic hosts.

Plant specialized metabolites are of great value to medicine and industry, but they are difficult to extract, and yields are low [133]. Research on the biosynthesis process of plant specialized metabolites provides a theoretical basis for improving or creating biological or chemical synthesis methods of specialized metabolites. Based on a specialized metabolite synthetic pathway, multiple promoter sequences are designed to precisely control and coordinate gene expression in the pathway, thereby reducing the accumulation of intermediate metabolites and cell load. For example, in *Saccharomyces cerevisiae*, a series of regulatable promoters with a length of no more than 100 bases was constructed by modularizing the promoters [134]. Synthetic biology studies of high-value medicinal specialized metabolites such as paclitaxel, vinblastine, and artemisinin using microbial chassis have been conducted [135,136]. Taking paclitaxel synthesis as an example, the reference genome sequence of *Taxus chinensis* can be used to comprehensively elucidate the paclitaxel biosynthesis pathway [137]. In synthetic biology, isopentenyl diphosphate is often used as a substrate to produce taxadiene in *E. coli*. In general, the biosynthesis of most natural products requires the interaction of cytochrome P450 and its reductase for electron transfer; however, it is difficult to express P450 in *E. coli* [138]. As *S. cerevisiae* has a rich endomembrane system in which cytochrome P450 can be efficiently functionally expressed, *E. coli* and *S. cerevisiae* can be used to produce paclitaxel using a co-culture method [139].

The cis elements recognized by MYC2 are used as promoters to drive all genes necessary for the synthesis of a natural product, and they are connected with JA-JAZ-MYC2 conditional molecular switches to form artificial chromosomes (Table 1). Because the promoter responds to JA and MYC2, a JA-JAZ-MYC2 conditional molecular switch includes the key elements of the JA signaling pathway, namely JAZ, COI1, MYC2, MED25, and other necessary genes. Transforming artificial chromosomes into chassis cells or substrate plants, large numbers of natural products can be synthesized by exogenous JA or exogenous stimulation (Figure 3). For example, the CDS of all necessary genes of the taxol synthesis pathway can be inserted into the promoter and terminator regulated by MYC2 separately (if the genes are activated by MYC2, their promoters can be used; if they are not regulated by MYC2, they have to be replaced with gene promoters activated by MYC2, such as *Dof2.1*, *ERF115*, etc.), and then connected to the JA-JAZ-MYC2 molecular switch for co-expression. In the application of synthetic biology, transcription factors and promoters that naturally exist in plants have been developed as molecular switches and promoters for efficient synthesis of specialized metabolites. For example, in rice, scientists have produced a construction containing eight anthocyanidin-related genes (two regulatory genes from maize and six structural genes from Perilla frutescens), which are driven by endosperm-specific promoters, to produce a new bio-enhanced germplasm “purple endosperm rice” (called “Zijingmi” in Chinese) with high anthocyanin content and antioxidant activity [140]. Researchers selected maize *ZmLc* (*Leaf color*) and *ZmPl* (*Purple leaf*), which encode the bHLH-type and MYB-type transcription factors, respectively, to activate the anthocyanin biosynthesis genes [141,142].

The chassis cells produced using plant metabolites include plant chassis and microbial cell chassis. Microbial cell chassis development utilizes *E. coli* and *S. cerevisiae* as typical hosts. *E. coli* is the most widely prokaryotic host used to produce heterogenous metabolites. As a chassis cell, it has the characteristics of rapid/easy growth, high product yield, and high cost efficiency. Further, the availability of various expression vectors and strains, the operation technology, and the relative ease of product purification make it an attractive host for industrial applications. However, the lack of cell intima as well as post-translational modification limit its use as a selection chassis for many plant natural products [143]. The use of plants as a specialized-metabolite production chassis has many advantages, including the presence of photosynthetic systems, extremely rich enzyme repertoires, and various cellular compartments [138,144,145]. Plant cellular compartments can decompose complete pathways into independent parts, and various conditions, such as reactions and precursor compounds, can be optimized in each cellular compartment; therefore, using plants as a chassis has great potential for production applications [126]. The commonly used plant chassis for heterologous synthesis is *Nicotiana benthamiana*, and various natural products, including alkaloid terpenoids, paclitaxel, and artemisinin, have transiently been expressed in *N. benthamiana* [146,147,148]. If higher plants are used as base plants, the JA-JAZ-MYC2 molecular switch can also be used to regulate the expression of MYC2 target genes in artificial chromosomes. As another synthetic biology chassis, microalgae have the advantages of having complex metabolic networks and biosynthesis pathways; in addition, they can be used as cell factories and have successfully synthesized metal nanoparticles [149,150,151]. If the base cell is a lower organism such as microalgae and has low homology with the genome of higher plants, it may not be able to start the expression of foreign artificial chromosomes; in such cases, additional JA-JAZ-MYC2 molecular switches need to be added to start the expression of artificial chromosomes.

The JA signal in plants is an important signal pathway in response to external stimuli. The inducibility and controllability of the JA-JAZ-MYC2 regulatory pathway endows it with the potential to be applied as a synthetic biological molecular switch. As a transcription factor, MYC2 itself can directly regulate the synthesis of a variety of plant specialized metabolites, and the promoter of its target gene can also be directly applied to the construction of artificial chromosomes regulated by the JA-JAZ-MYC2 molecular switch. Transgene expression only needs to occur at a certain point or time period in order to minimize the metabolic burden on the host cell, or to control the timing of gene expression. For this reason, unlike constitutive promoters such as *CaMV35S*, *Nos*, and *Ocs*, which are always active, the inducible gene expression system constructed by MYC2 can more dynamically and precisely regulate gene expression for optimal production of valuable chemicals [152]. An ideal control system should allow rapid and precise regulation of a target gene between the “ON” and “OFF” states or even simultaneous switching of different genes to the ON or OFF state [153,154,155]. Compatible “ON” and “OFF” switch functions for controlling the expression of genes in biosynthetic pathways and regulatory networks can also be achieved through rational design of MYC2-based switches. Therefore, MYC2 has some inherent advantages to be developed as a molecular switch for the synthesis of plant specialized metabolites via synthetic biology. This review synthesizes the role of MYC2 in the regulation of plant physiological processes and in the synthesis of specialized metabolites, summarizes the relevant pathways and genes related to MYC2, and provides new insights into and strategies for the application of MYC2 to synthetic biology to synthesize natural products with medicinal value.

## Figures and Tables

**Figure 1 ijms-24-03511-f001:**
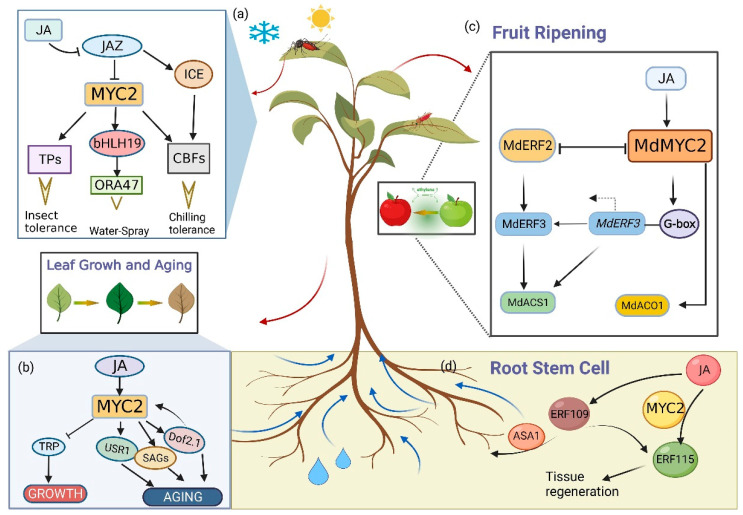
MYC2-mediated plant stress response to biotic/abiotic stress. (**a**) MYC2 mediates the response of plant leaves to insects, temperature, water spray, and other stresses by regulating genes such as *TPs*, *bHLH19*, and *CBFs*. (**b**) MYC2 regulates leaf vein development and leaf senescence through transcriptional regulation of *TRP*, *USR1*, *SAGs*, *Dof2.1,* and other genes. (**c**) During fruit ripening, MYC2 can regulate *ACS1*, *ACO1,* and other genes to participate in the regulation of ethylene synthesis. (**d**) MYC2 participates in root development by regulating genes such as *ERF115* and *ERF109*.

**Figure 2 ijms-24-03511-f002:**
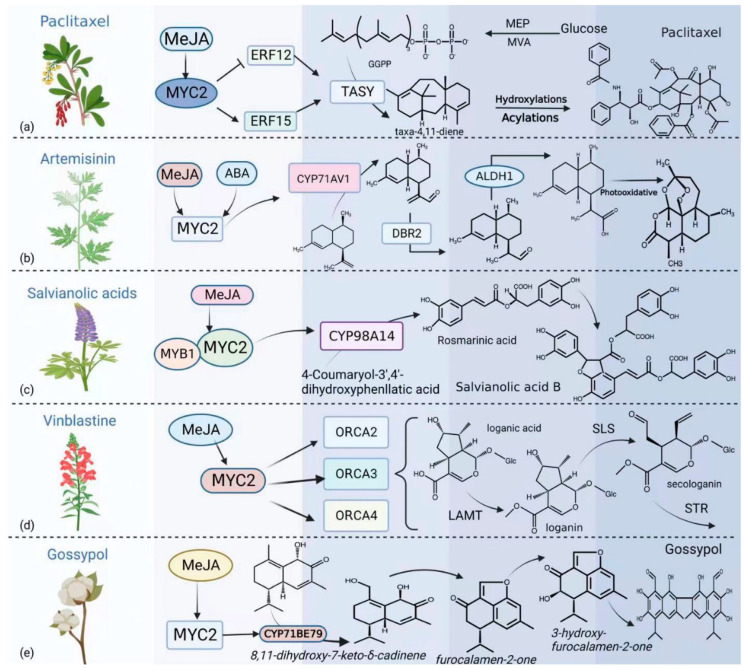
MYC2 is involved in the synthesis of plant specialized metabolites. (**a**) MYC2 is involved in the regulation of paclitaxel biosynthesis by regulating ERF12 and ERF15. (**b**) MYC2 can regulate the expression of key enzymes involved in artemisinin biosynthesis, such as CYP71AV1, AaDBR2, and AaALDH1. (**c**) MYC2 regulates the expression of *CYP98A14* to regulate the synthesis of *Salvia miltiorrhiza* specialized metabolites. (**d**) MYC2 affects vinblastine biosynthesis by regulating the transcription of ORCA2, ORCA3, and ORCA4. (**e**) MYC2 can directly regulate the expression of key enzymes in the gossypol synthesis pathway, such as CYP71BE79, and participate in the regulation of gossypol synthesis.

**Figure 3 ijms-24-03511-f003:**
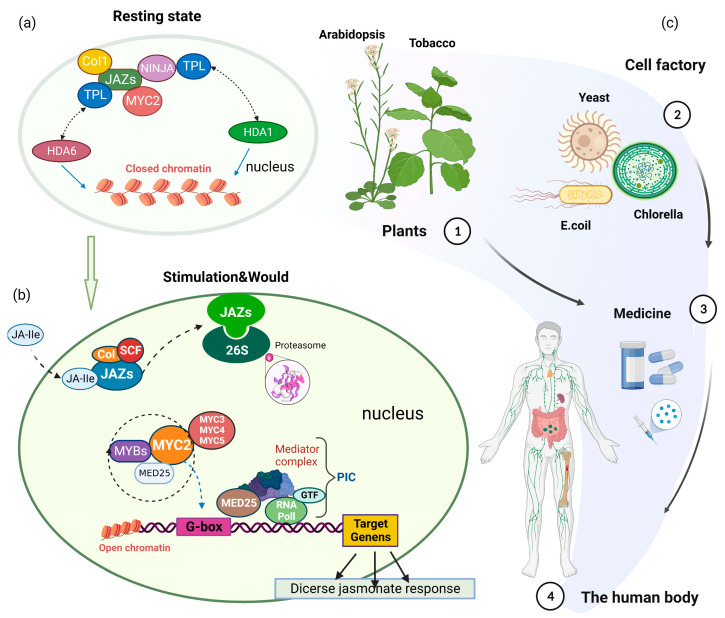
The central role of MYC2 in the JA signaling pathway and its application prospects in synthetic biology. (**a**) In the resting state, JAZ binds to and inhibits the transcription factor activity of MYC2 and inhibits the transcription of the JA gene. (**b**) Under stress, JA-Ile promotes the ubiquitination and degradation of JAZ, relieves the inhibition of MYC2 by JAZ, and activates the JA signaling pathway. (**c**) Applying the on–off function of MYC2 to synthetic biology techniques to produce medicinally valuable plant specialized metabolites using plants or microorganisms as chassis cells.

**Table 1 ijms-24-03511-t001:** Genes involved in MYC2 regulation.

Type of Function		List of Genes	Reference
Biotic and Abiotic Stress Conditions	Insect tolerance	*TPs* *MGAIs* *SGA* *PPC* *PR* *PDF1.2*	[59] [58] [64,65] [64,65] [67] [68]
	Water spray	*bHLH19* *ERF109* *ORA47*	[71] [71] [71]
	Drought stress	*rd22* *ADH*	[73] [73]
	Chilling tolerance	*MdCBF1*	[78]
Plant Growth and Development	Leaf growth and aging	*Dof2.1* *SAG12/13/29/113* *CAB1* *RBCS* *AtUSR1* *AMY3* *BAM1* *SUTI/4* *SWEET11* *SIPAO*	[82] [82,83] [82,83] [82,83] [85] [86] [86] [86] [86] [87]
	Root stem cell	*PLT1/2* *ERF115/109*	[88] [89]
	Fruit ripening and pollen maturation	*MdERF2/3* *MdACO1* *AGPs* *SS2* *BAM1*	[95] [94] [103] [103] [103]
Specialized Metabolites	Paclitaxel biosynthesis	*TcERF12/15*	[111]
	Artemisinin synthesis	*CYP71AV1* *DBR2*	[112,113] [112,113]
	Salvia miltiorrhiza	*CYP98A14* *SmGGPP*	[115] [115]
	Vinblastine biosynthesis	*ORCA2/3/4*	[116]
	Gossypol synthesis	*CYP71BE79*	[118]
	Psammosilene tunicoides synthesis	*TwTPS27a/b*	[119]

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
