# Peer review of "MYC2: A Master Switch for Plant Physiological Processes and Specialized Metabolite Synthesis"

_ijms, 2023, doi:10.3390/ijms24043511_

Round 1

Reviewer 1 Report

Comments to the Author

The bHLH transcription factor MYC, specially MYC2 plays an important role in plant development, defense, and specialized metabolite biosynthesis. Here the authors tried to summarize the role of MYC transcription factors in plants and proposed to use the MYC- switched JA signaling pathway in the synthetic biology approach to produce bioactive valuable metabolites.This review is interesting and has reference value. There are still some details that need to be further improved.

Comment 1: The role of MYC2, especially in salt tolerance is not discussed.

Comment 2: It is also known that the JA-MYC2 signaling pathway participates in the complex hormone network of plants, which controls numerous aspects of plant growth and development, as well as the way plants adapt to the environment. At the same time, MYC2 interacts with ABA, and ethylene (ET), and the crosstalk/interaction of GA and SA mediates various plant developmental processes and defense responses, please discuss.

Comment 3: LUH promotes JA-dependent enhancement of protein interactions between MYC2 

Comment 4: The role of the E3 ubiquitin ligase of CUL3 in the ubiquitination of MYC proteins. Please discuss.

Comment 5: In line 20 in the manuscript, pay attention to the singular and plural forms of roles.

Comment 6: Transcription factor ethylene response factor (ERF) 115 and line 198 of the manuscript are in the same format

Comment 7 

Figure 1: ICE and other font sizes are consistent with CBFs (please keep the fonts consistent in each figure, if there are important points, you can highlight them, but other suggestions should be consistent). 

Figure 2: It is recommended to keep the name written on each plant in the picture on the left consistent, for example, only write Cotton (plant name, if Latin is used in Latin, if English is in English) or both write cotton gossypol (plant name + secondary metabolites) name). 

Comment 8: 

The legends in Figures 1 and Figure 3 are suggested to be changed from MYC to MYC2.

Comment 9: What does "specialized metabolites" mean? Secondary metabolites, perhaps?

Comment 10: Please attach a table to summarize the genes involved in MYC2 regulation.

Author Response

Response to Reviewer 1 Comments

Dear Reviewers:

Thank you for your comments on our manuscript entitled "MYC2: A Master Switch for Plant Physiological Processes and Specialized Metabolite Synthesis" (ijms-2099327). Those comments were very helpful for improving our manuscript. We believed we have answered reviewer’s all comments. The main corrections in the paper and the responds to the reviewer’s comments are as follows:

Comments to the Author

The bHLH transcription factor MYC, specially MYC2 plays an important role in plant development, defense, and specialized metabolite biosynthesis. Here the authors tried to summarize the role of MYC transcription factors in plants and proposed to use the MYC- switched JA signaling pathway in the synthetic biology approach to produce bioactive valuable metabolites. This review is interesting and has reference value. There are still some details that need to be further improved.

Response: Thank you for your comments and suggestions, and we have carefully revised the entire manuscript and answered reviewer’s all comments.

Comment 1: The role of MYC2, especially in salt tolerance is not discussed.

Response 1: Thank you for your comment. In the manuscript, we supplemented the relevant content of MYC2 regulating salt stress. We have added “MYC2-like can mediate the expression of OsCYP2, a cyclophilin chaperone. Further, OsCYP2 has been proved to promote the resistance to abiotic stress when overexpressed, especially salt stress. In addition, MYC2-like is also a potential regulator in rice to regulate physiological processes related to salt stress through the abscisic acid (ABA) signaling pathway. A MYC2-like transcription factor binding to ABRE was also identified by yeast one hybrid assay and EMSA. An overexpression transformant of this transcription factor showed higher antioxidant enzyme activity in reactive oxygen species clearance. It is speculated that MYC2-like can improve the resistance to salt stress by improving the antioxidant enzyme activity and post-translational regulation of transformed plants [74]. in the paragraph 2 of section 3 (L235-L244).

Comment 2: It is also known that the JA-MYC2 signaling pathway participates in the complex hormone network of plants, which controls numerous aspects of plant growth and development, as well as the way plants adapt to the environment. At the same time, MYC2 interacts with ABA, and ethylene (ET), and the crosstalk/interaction of GA and SA mediates various plant developmental processes and defense responses, please discuss.

Response 2: Thank you for your comment. In the manuscript, we have discussed this part. We have added “The crosstalk/interaction between MYC2 and abscisic acid (ABA), ethylene (ET), GA, and SA mediate various plant developmental processes and defense responses. The signal transduction of ABA and JA is connected through the direct interaction between the ABA receptor PYL6 (RCAR9) and MYC2 [34]. In the ET signaling pathway, the interaction between MYC2 and EIN3 regulates the antagonism of JA and ET signals. MYC2 inhibits the transcriptional activity of EIN3 and EIL1 according to pull-down experiments. It also inhibited the expression of wound response genes and herbivore-induced genes induced by JA by interacting with EIN3 and inhibiting its DNA binding activity, and alleviated the defense of plants regulated by JA against herbivores [35]. The DELLA protein is a key inhibitor of the GA signal pathway, and it can directly interact with JAZ1, prevent JAZ1 from interacting with MYC2, and enhance the binding of MYC2 to its target gene promoter to promote JA reaction [36]. The SA and JA signaling pathways are known to intersect at various points because the two regulate biotic stress responses antagonistically [37]. Studies have shown NPR1 (which can activate SA response genes) to be a key player in the antagonistic crosstalk between SA and JA. The SA-facilitated suppression of JA-responsive genes, such as LIPOXYGENASE 2 (LOX2), VEGETATIVE STORAGE PROTEIN (VSP), and PDF1.2, was abolished in npr1 mutant plants [38]. MYC2 can participate in the regulation of the crosstalk between the SA and JA pathways. In the presence of SA, NPR1can interact with JA-induced MYC2 and inhibit the transcriptional activation of downstream response genes by interrupting the interaction be-tween MYC2 and MED25 [39].” in the paragraph 2 of section 2 (L125-L146).

Comment 3: LUH promotes JA-dependent enhancement of protein interactions between MYC2.

Response 3: Thank you for your comment. In the manuscript, we have discussed this part. We have added “LUH activates MYC2-directed transcription of JAZ2 and LOX2 via the mediator complex coactivator and the histone acetyltransferase HAC1. We showed that the mediator subunit MED25 physically recruits LUH to MYC2-target promoters, thereby linking MYC2 to HAC1-dependent acetylation of Lys-9 of histone H3 (H3K9ac) to activate JAZ2 and LOX2. LUH interacts with MED25 and HAC1 through its distinct domains, thus imposing a selective advantage by acting as a scaffold for MYC2 activation [29].” in paragraph 1 of section 2 (L119-L124).

Comment 4: The role of the E3 ubiquitin ligase of CUL3 in the ubiquitination of MYC proteins. Please discuss.

Response 4: Thank you for your comment. In the manuscript, we have discussed this part. We have added “Inhibition of MYC activity is therefore necessary to reset JA signaling and avoid deleterious runaway responses.” in paragraph 3 of section 2 (L168-L169).

We have added “In addition, E3 ubiquitin ligase can also reduce MYC protein level based on Cullin3 and BPM proteins as substrate junctions. JA enhances BPM3 stability and establishes a negative feedback regulatory loop to control MYC level and activity. In this new JA pathway regulatory layer, MYC is degraded and terminated by CUL3 BPM-mediated MYC activity [45].” in paragraph 3 of section 2 (L175-L179).

Comment 5: In line 20 in the manuscript, pay attention to the singular and plural forms of roles.

Response 5: Thank you for your comment. We have modified “defense” to “defenses” (L20).

Comment 6: Transcription factor ethylene response factor (ERF) 115 and line 198 of the manuscript are in the same format

Response 6: Thank you for your comment. We have modified “Transcription factor ethylene response factor (ERF) 115” to “The transcription factor ethylene response factor 115 (ERF115)” (L292).

Comment 7: Figure 1: ICE and other font sizes are consistent with CBFs (please keep the fonts consistent in each figure, if there are important points, you can highlight them, but other suggestions should be consistent). Figure 2: It is recommended to keep the name written on each plant in the picture on the left consistent, for example, only write Cotton (plant name, if Latin is used in Latin, if English is in English) or both write cotton gossypol (plant name + secondary metabolites) name).

Response 7: Thank you for your comment. We have modified Figure 1 and Figure 2 according to your suggestions. The new pictures have been uploaded again.

Comment 8: The legends in Figures 1 and Figure 3 are suggested to be changed from MYC to MYC2.

Response 8: Thank you for your comment. We have modified “MYC” to “MYC2” in the legends of Figures 1 and Figure 3.

Comment 9: What does "specialized metabolites" mean? Secondary metabolites, perhaps?

Response 9: Thank you for your comment. Secondary metabolites, also called specialized metabolites, toxins, secondary products. Bacteria, fungi, or plants, which are not directly involved in the normal growth, development, or reproduction of the organism. Instead, they generally mediate ecological interactions, which may produce a selective advantage for the organism by increasing its survivability or fecundity. Specific secondary metabolites are often restricted to a narrow set of species within a phylogenetic group. Secondary metabolites often play an important role in plant defense against herbivory and other interspecies defenses. Humans use secondary metabolites as medicines, flavourings, pigments, and recreational drugs.

Specialized plant metabolites (SPMs), traditionally referred to as 'secondary metabolites', are chemical compounds involved in a broad range of biological functions, including plant responses to abiotic and biotic stresses. Moreover, some of them have a role in end-product quality with potential health benefits in humans. For this reason, they became an important target of studies focusing on their mechanisms of action and use in crop breeding and management (Marone, Mastrangelo et al. 2022).

Reference:

Marone, D., A. M. Mastrangelo, G. M. Borrelli, A. Mores, G. Laido, M. A. Russo and D. B. M. Ficco (2022). "Specialized metabolites: Physiological and biochemical role in stress resistance, strategies to improve their accumulation, and new applications in crop breeding and management." Plant Physiol Biochem 172: 48-55.

Comment 10: Please attach a table to summarize the genes involved in MYC2 regulation.

Response 10: Thank you for your comment. We have attached a table to summarize the genes involved in MYC2 regulation.

Table 1 Genes involved in MYC2 regulation

Type of function

List of genes

Reference

Biotic and Abiotic Stress Conditions

Insects tolerance

TPs

MGAIs

SGA

PPC

PR

PDF1.2

[58]

[57]

[63,64]

[63,64]

[66]

[67]

Water sprsy

bHLH19

ERF109

ORA47

[70]

[70]

[70]

Drought stress

rd22

ADH

[72]

[72]

Chilling tolerance

MdCBF1

[77]

Plant Growth and Development

Leaf growth and aging

Dof2.1

SAG12/13/29/113

CAB1

RBCS

AtUSR1

AMY3

BAM1

SUTI/4

SWEET11

SIPAO

[81]

[82,83]

[82,83]

[82,83]

[84]

[85]

[85]

[85]

[85]

[86]

Root stem cell

PLT1/2

ERF115/109

[87]

[88]

Fruit ripening and

MdERF2/3

MdACO1

AGPs

SS2

BAM1

[94]

[93]

[102]

[102]

[102]

specialized metabolites

Paclitaxel biosynthesis

TcERF12/15

[110]

Artemisinin synthesis

CYP71AV1

DBR2

[111,112]

[111,112]

Salvia miltiorrhiza.

CYP98A14

SmGGPP

[114]

[114]

Vinblastine biosynthesis

ORCA2/3/4

[115]

Gossypol synthesis

CYP71BE79

[117]

Psammosilene tunicoides synthesis

TwTPS27a/b

[118]

Reviewer 2 Report

This view is about the master role of MYC2 in regulating plant development and defense, this paper is well written, however, there are many grammatical mistakes, and many references cited in this paper were not properly labled, ex. Line 248-249. Thus, the manuscript should be carefully revised by a native speaker. Below are my main concerns:

1, I think the first paragraph is improper, since this paper is about how JA signaling pathway/MYC2 modulate plant specialized metabolite synthesis, not about the medical function of specialized metabolites.

2, Line 109-140, I think this paragraph is disordered, the authors should combine insect pests and pathogens together, not for two sentences.

3, The JA signaling pathway is the most important pathway in modulating plant defense to insect herbivores, however, I did not see any typical (widely known) examples.

4, In fact, JA-Ile is the bioactive molecules of JA receptor, so in line 52-54, I think the statement is improper.

5, In the part ‘MYC2 is Involved in the Regulation of Plant Growth and Development’, most of examples were listed from Arabidopsis, I am wondering if the importance of MYC2 in regulating plant growth and development widely exists in plant kingdom or just in Arabidopsis. As I know, OsMYC2 participates spikelet development in rice (Cai et al., 2014). So I think this part should revised and examples from other plants should be included.

Cai et al, 2014. Jasmonic acid regulates spikelet development in rice. Nature Communications.

6, Line 239, MYC2 is not the effector of JA signaling.

7, This paper wants to say that the MYC2 functions as a master switch for plant physiological processes and specialized metabolite synthesis, however, I did not see any connection between plant physiological processes and specialized metabolite synthesis. The authors just list some examples in these two processes.

The authors should consider the following suggestions:

Line 35, ‘flee from’.

Line 36-38, the production of specialized metabolites is induced by stresses, not only injury.

Line 45, ‘involves in’.

Line 75, ‘SCFCOI1’.

Line 130, 134 the Scientific name of tomato, brown planthopper.

Line 129, ‘Fusarium wilt’.

Line 241, ‘involving’.

Line 248, ‘Overexpression of TcMYC2a’.

Line 252, ‘AaMYC2, AaNAC1, and AaHD1’.

Line 270-272, which kind of plant?  Please check the whole MS.

The reference should be checked thoroughly, ex., ref. 56.

Author Response

Response to Reviewer 2 Comments

Dear Reviewers:

Thank you for your comments on our manuscript entitled "MYC2: A Master Switch for Plant Physiological Processes and Specialized Metabolite Synthesis" (ijms-2099327). Those comments were very helpful for improving our manuscript. We believed we have answered reviewer’s all comments. The main corrections in the paper and the responds to the reviewer’s comments are as follows:

Comments to the Author

This view is about the master role of MYC2 in regulating plant development and defense, this paper is well written, however, there are many grammatical mistakes, and many references cited in this paper were not properly labled, ex. Line 248-249. Thus, the manuscript should be carefully revised by a native speaker.

Responses: Thank you for your comment. We proofread all the references cited in this paper and correct the inaccurate references. The revised references are highlighted in yellow.

For example, in line 248-249, we have modified “TcMYC2 activates paclitaxel biosynthesis in response to MeJA and binds the promoter of the paclitaxel biosynthesis gene encoding taxadiene synthase [76]. TcMYC2a overexpression increases the expression of TcERF12 and TcERF15 to 248 regulate taxadiene synthase gene expression.” To “TcMYC2 activates paclitaxel biosynthesis in response to MeJA and binds to the pro-moter of the paclitaxel biosynthesis gene encoding taxadiene synthase. TcMYC2a overexpression increases the expression of TcERF12 and TcERF15 to regulate taxadiene synthase gene expression [110].” in the paragraph 1 of section 5 (L376-L379).

We have modified “Compared with WT plants, Arabidopsis myc2 mutants have higher auxin contents and a significantly increased leaf vein density [4].” to “Compared with WT plants, Arabidopsis myc2 mutants have higher auxin contents and a significantly increased leaf vein density [80].” in the paragraph 1 of section 4 (L266-L267).

We have modified “MYC2 directly regulates aureotriterpenoid synthesis by regulating the backbone-building enzymes β-amyrin synthase and squalene epoxidase in the aureotriterpenoid saponins synthesis pathway [85].” to “MYC2 directly regulates the synthesis of triterpenoid saponins of Psammosilene tunicoides synthesis by regulating the backbone-building enzymes β-amyrin synthase and squalene epoxidase in the triterpenoid saponin synthesis pathway [119].” in the paragraph 2 of section 5 (L400-L403).

At the same time, we have carefully revised the manuscript by native speakers (Editage) to avoid grammatical errors. The proof of English language editing is shown in the figure below.

Reference

  1. Huang, C.F.; Yu, C.P.; Wu, Y.H.; Lu, M.Y.J.; Tu, S.L.; Wu, S.H.; Shiu, S.H.; Ku, M.S.B.; Li, W.H. Elevated Auxin Biosynthesis and Transport Underlie High Vein Density in C4 Leaves. Proc. Natl. Acad. Sci. U. S. A. 2017, 114, E6884–E6891, doi:10.1073/pnas.1709171114.
  2. Zhang, M.; Jin, X.; Chen, Y.; Wei, M.; Liao, W.; Zhao, S.; Fu, C.; Yu, L. TcMYC2a, a Basic Helix–Loop–Helix Transcription Factor, Transduces JA-Signals and Regulates Taxol Biosynthesis in Taxus Chinensis. Front. Plant Sci. 2018, 9, 1–13, doi:10.3389/fpls.2018.00863.
  3. Ribeiro, B.; Lacchini, E.; Bicalho, K.U.; Mertens, J.; Arendt, P.; Bossche, R. Vanden; Calegario, G.; Gryffroy, L.; Ceulemans, E.; Buitink, J.; et al. A Seed-Specific Regulator of Triterpene Saponin Biosynthesis in Medicago Truncatula. Plant Cell 2020, 32, 2020–2042, doi:10.1105/tpc.19.00609.

Below are my main concerns:

Comment 1: I think the first paragraph is improper, since this paper is about how JA signaling pathway/MYC2 modulate plant specialized metabolite synthesis, not about the medical function of specialized metabolites.

Response 1: Thank you for your suggestion, which is very useful. After adjusting the manuscript according to your suggestion, the logic of the manuscript have been clearer and smoother. We have replaced the first and second paragraphs of the first part.

In fact, according to previous research reports, we all know that MYC2, as the main regulator of the JA signaling pathway, participates in the regulation of plant physiological processes and the synthesis of special metabolites. In particularly, it plays an important role in the synthesis of secondary metabolites. However, the underlying mechanism of how to use MYC2 to regulate the synthesis of specialized metabolites in plants has not been described, so we consider the use of synthetic biology to design MYC2-driven chassis cells for the synthesis of specialized metabolites with high medicinal value, such as paclitaxel, vincristine, and artemisinin, seem to be a promising strategy, so this section is also important. We put this part in the second paragraph of the first part.

Comment 2: Line 109-140, I think this paragraph is disordered, the authors should combine insect pests and pathogens together, not for two sentences.

Response 2: Thank you for your comment. We have confused MYC's regulation of insect resistance and disease resistance, which really makes it difficult for readers to read, so we have modified and moved the description of disease resistance “Similarly, in tomato, magnesium oxide positively regulates resistance genes by triggering JA signaling and activating MYC2 to induce immunity to Fusarium wilt [37].” in the paragraph 1 of section 3 (L127-L129) to “In addition, MYC is involved in regulating the expression of disease resistance related genes to regulate plant resistance to diseases. In Solanum lycopersicum, magnesium oxide positively regulates resistance genes by triggering JA signaling and activating MYC2 to induce immunity to Fusarium wilt [65].” in the paragraph 1 of section 3 (L215-L219). Because there are few descriptions of disease resistance in the manuscript, we added the example of MYC regulating disease resistance mechanism “Overexpression of OsMYC2 induces up-regulation of PR gene and bacterial blight re-sistance in rice [66]. At the same time, MYC2, as a negative regulator of plant pathogens, can block the function of EIN3 by reducing the ET-mediated pathogen reaction. Further, MYC2 can interact with EIN3 to antagonize and regulate the expression of ERF, thereby enhancing the expression of PDF1.2 induced by pathogens [67]." in the paragraph 1 of section 3 (L219-L223).

Comment 3: The JA signaling pathway is the most important pathway in modulating plant defense to insect herbivores, however, I did not see any typical (widely known) examples.

Response 3: Thank you for your comment. Plants sit at the bottom of the food chain and are constantly under attack by herbivores, many of which are insects. To survive, plants have evolved complex mechanisms to defend against these attackers. In the 1 paragraph of section 3 of the manuscript (L186-L192), we have added “Phytohormone signaling pathways mediated by JA play a key role in many of these defense responses. When attacked by herbivores, plants activate a series of synthetases to trigger JA biosynthesis, including 13-lipoxygenase (13-LOX), AOS, oxyalkylene, cyclase and OPR3 [51–53]. This in turn triggers a series of JA-dependent downstream reactions, such as biosynthesis of several metabolites, and molecules, such as the de-fense protein thionine (encoded by THI) and nutrient storage protein 1 (encoded by VSP1) [54,55].

Comment 4: In fact, JA-Ile is the bioactive molecules of JA receptor, so in line 52-54, I think the statement is improper.

Response 4: Thank you for your comment. We have revised the sentence in lines 52-54, and modified “After plant damage, a large amount of JA is produced to activate MYC2 by degrading JAZ (Jasmonate ZIM-domain), thereby activating transcription of downstream genes of MYC2, thus triggering plant defense and repair [7].” to “After the plant is damaged, a large amount of JA is produced, which is then transformed into (+)-7-iso-jasmonoyl-L-isoleucine (JA-Ile) with biological activity. JA-Ile promotes the formation of SCF-COI1-JAZ coreceptor complex, degrades JAZ (Jasmonate ZIM-domain) through 26S proteasome ubiquitination, relieves JAZ's inhibition of MYC2, thereby activating the transcription of MYC2 downstream genes, thus triggering plant defense and repair [3].” in the paragraph 1 of section 1 (L39-L44).

Comment 5: In the part ‘MYC2 is Involved in the Regulation of Plant Growth and Development’, most of examples were listed from Arabidopsis, I am wondering if the importance of MYC2 in regulating plant growth and development widely exists in plant kingdom or just in Arabidopsis. As I know, OsMYC2 participates spikelet development in rice (Cai et al., 2014). So I think this part should revised and examples from other plants should be included.

Response 5: Thank you for your comment. The importance of MYC2 in regulating plant growth and development does not only exist in Arabidopsis, but also widely exists in the plant kingdom. This part should indeed include examples of other plants.

We have added “MeJA treatment and SlMYC2 overexpression inhibited Solanum lycopersicum seedling growth and photosynthesis, but increased the sugar acid ratio and content of lycopene, carotenoid, soluble sugar, total phenol, and flavone, indicating that JA signal transduction could inhibit Solanum lycopersicum seedling growth and change fruit quality [9].” in the 3 paragraph of section 4 of the manuscript (L319-L323).

We have added “The spikelet is the basal unit of inflorescence in grasses, and its formation is crucial for reproductive success and cereal yield. A study found that JA plays a key role in deter-mining rice (Oryza sativa) spikelet morphogenesis. Extra glume 1 (eg1) and eg2 mutants exhibit altered spikelet morphology, with changed floral organ identity and number, as well as defective floral meristem determinacy. EG1 is a plastid-targeted lipase that participates in JA biosynthesis, and EG2/OsJAZ1 is a JA signaling repressor that interacts with a putative JA receptor, OsCOI1b, to trigger OsJAZ1 degradation during spikelet development. OsJAZ1 also interacts with OsMYC2 and represses OsMYC2 role in activating OsMADS1, an E-class gene crucial for spikelet development [101]. NtMYC2a plays an important role in carbohydrate metabolism and pollen development by regulating the expression of the starch metabolism-related genes AGPs, SS2, and BAM1 in pollen grains, anther walls and ovaries of tobacco plants. The process of pollen maturation was accelerated in NtMYC2a-OE plants and was delayed in NtMYC2a-RI plants [102]. Chestnut (Castanea mollisima) is an important woody food crop, but its yield is low when cultivated, mainly due to the problems of fewer female flowers and more male flowers. A higher concentration of JA-Ile is conducive for the differentiation and formation of female flower buds during post-winter, and JAZ1-3 and MYC2-1 play a key role in the differentiation of female flower buds of chestnut [103].” in the 4 paragraph of section 4 of the manuscript (L342-L360).

Comment 6: Line 239, MYC2 is not the effector of JA signaling.

Response 6: Thank you for your comment. MYC2 is a master regulator of JA signaling. We have revised the sentence in line 239, and modified “As the main effector of JA signaling” to “As the main regulator of JA signaling” in the 1 paragraph of section 5 of the manuscript (L369).

Comment 7: This paper wants to say that the MYC2 functions as a master switch for plant physiological processes and specialized metabolite synthesis, however, I did not see any connection between plant physiological processes and specialized metabolite synthesis. The authors just list some examples in these two processes.

Response 7: Thank you for your comment. There is a relationship between plant physiological processes and the synthesis of specialized metabolites. Plants can resist different biological stresses by releasing secondary metabolites to regulate immune response. For example, in the 1 paragraph of part 3 (L198-203) of the manuscript, we wrote “Similarly, MYC-related mutants reportedly show differential resistance to cotton bollworm (Helicoverpa armigera), which may be related to the roles of MGAIs (MYC-related genes against insects) in flavonoid biosynthesis [57]. High concentrations of flavonoids inhibit the growth of certain insects and allow plants to recover after in-jury or insect attack. MYC2 may regulate flavonoid production by acting on MGAIs to alleviate plant growth inhibition caused by insect damage [57].”. In the 1 paragraph of part 3 (L211-215) of the manuscript, we wrote “Members of the Solanaceae family accumulate phenylpropanoid-polyamine conjugates (PPCs) in response to attackers while also maintaining a chemical barrier of steroidal glycoalkaloids (SGAs). Solanum lycopersicum MYC1 and MYC2, redundantly control JA-inducible PPC and SGA production, and are also essential for constitutive SGA biosynthesis [63,64].”.

The authors should consider the following suggestions:

Comment 8: Line 35, ‘flee from’.

Response 8: Thank you for your suggestion. We have modified “flee biotic or abiotic stress” to “flee from biotic or abiotic stress” (L51).

Comment 9: Line 36-38, the production of specialized metabolites is induced by stresses, not only injury.

Response 9: Thank you for your suggestion. We have modified “Instead, the production of specialized metabolites is a way in which plants respond to injury; indeed, when damaged, plants generate and accumulate specialized metabolites to improve their immunity and resistance” to “Plants therefore produce specialized metabolites to respond to stress; indeed, when subjected to stress, plants generate and accumulate specialized metabolites to improve their immunity and resistance [13].” in the 2 paragraph of section 1 of the manuscript (L52-54).

Comment 10: Line 45, ‘involves in’.

Response 10: Thank you for your suggestion. We have modified “involves cumbersome steps” to “involves in cumbersome steps” in the 2 paragraph of section 1 of the manuscript (L65).

Comment 11: Line 75, ‘SCFCOI1’.

Response 11: Thank you for your suggestion. We have modified “SCFCOI1” to “SCFCOI1” in the 1 paragraph of section 2 of the manuscript (L84).

Comment 12: Line 130, 134 the Scientific name of tomato, brown planthopper.

Response 12: Thank you for your suggestion. We have modified “tomato” to “Solanum lycopersicum” (L205, L206, L213, L217). We have modified “brown planthopper” to “Nilaparvata lugens” (L210).

Comment 13: Line 129, ‘Fusarium wilt’.

Response 13: Thank you for your suggestion. We have modified “Fusarium wilt” to “Fusarium wilt” in the 1 paragraph of section 3 of the manuscript (L218-219).

Comment 14: Line 241, ‘involving’.

Response 14: Thank you for your suggestion. We have modified “involved” to “involving” in the 1 paragraph of section 5 of the manuscript (L370).

Comment 15: Line 248, ‘Overexpression of TcMYC2a’.

Response 15: Thank you for your suggestion. We have modified “TcMYC2a overexpression” to “Overexpression of TcMYC2a” in the 1 paragraph of section 5 of the manuscript (L378).

Comment 16: Line 252, ‘AaMYC2, AaNAC1, and AaHD1’.

Response 16: Thank you for your suggestion. We have modified “AaMYC2, AaNAC1, and AaHD1” to “AaMYC2, AaNAC1, and AaHD1” in the 1 paragraph of section 5 of the manuscript (L382-383).

Comment 17: Line 270-272, which kind of plant?  Please check the whole MS.

Response 17: Thank you for your suggestion. The plant is Psammosilene tunicoides. We have modified “aureotriterpenoid” to “triterpenoid saponin of Psammosilene tunicoides” in the 2 paragraph of section 5 of the manuscript (L401).

Comment 18: The reference should be checked thoroughly, ex., ref. 56.

Response 18: Thank you for your suggestion. We proofread all the references cited in this manuscript and correct the inaccurate references.

We have modified “In senescent leaves, JA induces H2O2 accumulation via binding of MYC2 to the CAT2 promoter to inhibit its expression, thus upregulating the expression of the senescence-related genes SAG12, SAG13, SAG29, and SAG113 and inhibiting (Figure 1b) that of the photosynthesis-related genes CAB1 and RBCS [55,56].” to “In senescent leaves, JA induces H2O2 accumulation via binding of MYC2 to the CAT2 promoter to inhibit its expression, thus upregulating the expression of the senescence-related genes SAG12, SAG13, SAG29, and SAG113 and inhibiting (Figure 1b) that of the photosynthesis-related genes CAB1 and RBCS [82,83].” in the paragraph 1 of section 4 (L270-L274).

We have modified “In apple, MdMYC2 interacts with the G-box in the MdCBF1 promoter to regulate freezing tolerance (Figure 1a) [51].” to “In apple, MdMYC2 interacts with the G-box in the MdCBF1 promoter to regulate freezing tolerance (Figure 1a) [77].” in the paragraph 2 of section 3 (L246-L247).

We have modified “In Arabidopsis, JA inhibits cell division in the main root meristem in a MYC2-dependent manner. MYC2 directly binds to the promoters of PLT1 and PLT2 to inhibit their expression and thus negatively regulates root stem cell-maintenance and root meristem activity [59].” to “In Arabidopsis, JA inhibits cell division in the main root meristem in a MYC2-dependent manner. MYC2 directly binds to the promoters of PLT1 and PLT2 to inhibit their expression and thus negatively regulates root stem cell-maintenance and root meristem activity [87].” in the paragraph 2 of section 4 (L289-L292).

Reference

  1. Wang, Y.; Xu, H.; Liu, W.; Wang, N.; Qu, C.; Jiang, S.; Fang, H.; Zhang, Z.; Chen, X. Methyl Jasmonate Enhances Apple’ Cold Tolerance through the JAZ–MYC2 Pathway. Plant Cell. Tissue Organ Cult. 2019, 136, 75–84, doi:10.1007/s11240-018-1493-7.
  2. Zhang, Y.; Ji, T.T.; Li, T.T.; Tian, Y.Y.; Wang, L.F.; Liu, W.C. Jasmonic Acid Promotes Leaf Senescence through MYC2-Mediated Repression of CATALASE2 Expression in Arabidopsis. Plant Sci. 2020, 299, 110604, doi:10.1016/j.plantsci.2020.110604.
  3. Qi, T.; Wang, J.; Huang, H.; Liu, B.; Gao, H.; Liu, Y.; Song, S.; Xie, D. Regulation of Jasmonate-Induced Leaf Senescence by Antagonism between BHLH Subgroup IIIe and IIId Factors in Arabidopsis. Plant Cell 2015, 27, 1634–1649, doi:10.1105/tpc.15.00110.
  4. Chen, Q.; Sun, J.; Zhai, Q.; Zhou, W.; Qi, L.; Xu, L.; Wang, B.; Chen, R.; Jiang, H.; Qi, J.; et al. The Basic Helix-Loop-Helix Transcription Factor Myc2 Directly Represses Plethora Expression during Jasmonate-Mediated Modulation of the Root Stem Cell Niche in Arabidopsis. Plant Cell 2011, 23, 3335–3352, doi:10.1105/tpc.111.089870.

Reviewer 3 Report

This review shows a comprehensive network of interactions of MYC2 transcription factor for plants metabolite biosynthesis. Pointing out the advantages of stimulating or repressing the expression of biosynthetic enzymes for metabolite profiles modulation and to produce molecules of economic interests.

There are some topics to be further explained to improve the contents of the manuscript.

1) At the introduction indicate the commonly used methods of metabolites extraction, for example: distillation, organic extractions, solid phase extraction.

2) What are the currently most needed metabolites which can be obtained from plants? Medicinal, precursors, nutritional, industrial molecules?

3) At the paragraph from line 48, you can include how volatile forms of JA are used as communication between neighboring plants for stimulating defense and metabolite production.

4) Does MYC2 is involved or crosstalk with other hormone pathways?

5) In section 2, the interactions between JA and salicylates can be included, since both hormones can modulate biotic and abiotic interactions and defense responses.

6) Line 81-82. What response enzymes are under the command of G-box elements? What is the abundance of JA-regulated enzymes which possesses G-box elements?

7) Line 89, what chromatin-modifying enzymes are regulated by MYC?

8) Line 104, what photoreceptors could be involved in the blue light sensing and MYC2 pathway? Is this phenomenon coordinated with photoperiod or circadian responses?

9) In section 3, What are the most synthesized terpenes with repellent/insecticide properties in Arabidopsis?

10) What are the most abundant flavonoids with insecticide/repellent properties in Helicoverpa?

11) Are PR proteins also regulated by MYC2 for positive interactions against fungi or insects?

12) Drought and freezing stress response both shares the osmolyte production response for liquid water retention.

13) In section 4, does MYC regulation of only nuclear encoded genes? Can MYC regulate plastidial genes?

14) In senescence, does MYC2 signaling activates the expression of transporters, ion pumps, degradation enzymes, cell wall remodeling enzymes?

15) Paragraph at line 188. Some root remodeling response involve JA and brassinosteroids for nitrogen and phosphate uptake.

16) Paragraph at line 202. Are there intertwined interactions between MYC2 with gibberellin-response genes for fruit maintenance and seed development? Or interactions with ethylene and ABA-response genes for fruit ripening?

17) MYC2-regulated morphogenesis of flower androcium occurs in dioecious/monoecious plants? Or hermaphrodite flowers? Or occurs for self-compatibility/incompatibility pollination?

18) In section 5, regarding bionsynthesis genes for paclitaxel production, are these mostly regulated by MYC2? Or are only branches of the biosynthesis pathway regulated by MYC2?

19) It seems MYC2 frequently regulates terpenoids biosynthesis pathways in all given examples.

20) Are there MYC2 regulated terpenoid biosynthesis for eucalyptus, pine or conifers?

21) In section 6, What characteristics should accomplish chassis cells for plant metabolite production? What are their advantages and limitations?

22) How many genes can be incorporated into an artificial chromosome?

23) How efficient are the post-translational modifications in Saccharomices gene expression systems?

24) What molecules have been successfully produced using microalgae expression systems?

Author Response

Response to Reviewer 3 Comments

Dear Reviewers:

Thank you for your comments on our manuscript entitled “MYC2: A Master Switch for Plant Physiological Processes and Specialized Metabolite Synthesis” (ijms-2099327). Those comments were very helpful for improving our manuscript. We believed we have answered reviewer’s all comments. The main corrections in the paper and the responds to the reviewer’s comments are as follows:

Comments to the Author

This review shows a comprehensive network of interactions of MYC2 transcription factor for plants metabolite biosynthesis. Pointing out the advantages of stimulating or repressing the expression of biosynthetic enzymes for metabolite profiles modulation and to produce molecules of economic interests.

There are some topics to be further explained to improve the contents of the manuscript.

Comment 1: At the introduction indicate the commonly used methods of metabolites extraction, for example: distillation, organic extractions, solid phase extraction.

Response 1: Thank you for your comment. We have replaced the “However, the extraction of plant specialized metabolites is limited by various factors, including low yields and long plant-growth cycles.” to “Secondary metabolites of plants can be extracted by distillation, organic extraction, solid phase extraction, super critical fluid extraction, pressurized liquid extraction and microwave assisted extraction, but they are restricted by the low yield and long plant growth cycle [17].” in the paragraph 2 of section 1 (L60-L63).

Comment 2: What are the currently most needed metabolites which can be obtained from plants? Medicinal, precursors, nutritional, industrial molecules?

Response 2: Thank you for your comment. We have replaced the “Plant specialized metabolites not only induce plant resistance to stress but have extremely high medicinal value to humans as well [3]” to “Plant specialized metabolites not only induce plant resistance to stress, but are also widely used in the chemical, food, and agriculture industriese, especially in the medical field. They can be divided into tannins, flavonoids, terpenoids, alkaloids, quinones, etc. At present, the most important metabolites from plants are taxol, vinblastine, camptothecin, and artemisinin [14,15].” in the paragraph 2 of section 1 (L54-L58).

Comment 3: At the paragraph from line 48, you can include how volatile forms of JA are used as communication between neighboring plants for stimulating defense and metabolite production.

Response 3: Thank you for your comment. In the paragraph 1 of part 1 (L47-L50), we have added “In addition, JA not only activate the defense response of the plant itself, but also enhance the content of linalool [10] and β-ocimene or induce the release of other volatile organic compounds, such as shikimic acid derivatives, to enhance the resistance of adjacent plants to the attack of arthropod herbivores [11].”.

Comment 4: Does MYC2 is involved or crosstalk with other hormone pathways?

Response 4: Thank you for your comment. MYC2 is crosstalk with other hormone pathways. In the manuscript, we have discussed this part. We have added “The crosstalk/interaction between MYC2 and abscisic acid (ABA), ethylene (ET), GA, and SA mediate various plant developmental processes and defense responses. The signal transduction of ABA and JA is connected through the direct interaction between the ABA receptor PYL6 (RCAR9) and MYC2 [34]. In the ET signaling pathway, the interaction between MYC2 and EIN3 regulates the antagonism of JA and ET signals. MYC2 inhibits the transcriptional activity of EIN3 and EIL1 according to pull-down experiments. It also inhibited the expression of wound response genes and herbivore-induced genes induced by JA by interacting with EIN3 and inhibiting its DNA binding activity, and alleviated the defense of plants regulated by JA against herbivores [35]. The DELLA protein is a key inhibitor of the GA signal pathway, and it can directly interact with JAZ1, prevent JAZ1 from interacting with MYC2, and enhance the binding of MYC2 to its target gene promoter to promote JA reaction [36]. The SA and JA signaling pathways are known to intersect at various points because the two regulate biotic stress responses antagonistically [37]. Studies have shown NPR1 (which can activate SA response genes) to be a key player in the antagonistic crosstalk between SA and JA. The SA-facilitated suppression of JA-responsive genes, such as LIPOXYGENASE 2 (LOX2), VEGETATIVE STORAGE PROTEIN (VSP), and PDF1.2, was abolished in npr1 mutant plants [38]. MYC2 can participate in the regulation of the crosstalk between the SA and JA pathways. In the presence of SA, NPR1 can interact with JA-induced MYC2 and inhibit the transcriptional activation of downstream response genes by interrupting the interaction be-tween MYC2 and MED25 [39].” in the paragraph 2 of section 2 (L125-L146).

Comment 5: In section 2, the interactions between JA and salicylates can be included, since both hormones can modulate biotic and abiotic interactions and defense responses.

Response 5: Thank you for your comment. In the manuscript, we have added the interactions between JA and salicylates. We have added “The SA and JA signaling pathways are known to intersect at various points because the two regulate biotic stress responses antagonistically [37]. Studies have shown NPR1 (which can activate SA response genes) to be a key player in the antagonistic crosstalk between SA and JA. The SA-facilitated suppression of JA-responsive genes, such as LIPOXYGENASE 2 (LOX2), VEGETATIVE STORAGE PROTEIN (VSP), and PDF1.2, was abolished in npr1 mutant plants [38]. MYC2 can participate in the regulation of the crosstalk between the SA and JA pathways. In the presence of SA, NPR1 can interact with JA-induced MYC2 and inhibit the transcriptional activation of downstream response genes by interrupting the interaction be-tween MYC2 and MED25 [39].” in the paragraph 2 of section 2 (L136-L146).

Comment 6: Line 81-82. What response enzymes are under the command of G-box elements? What is the abundance of JA-regulated enzymes which possesses G-box elements?

Response 6: Thank you for your comment. G-box (CACGTG) is a DNA element widely found in the plant genome, almost all of which are regulated by enzymes such as the transcription factor family bZIP (leucine zipper), bHLH (helix-loop-helix), and ABA-responsive transcription factor G-class bZIP TF2 (GBF2) (Ko and Brandizzi, 2022). In addition, Aldo-keto reductases (AKRs) also contain G-box sequences (Suekawa et al., 2018). Because our article is mainly about MYC2, which is a key positive regulator of the JA signaling pathway, I searched for genes regulated by JA through the Tair website (https://www.arabidopsis.org/), screened for enzymes containing G-boxes.

We added “In Arabidopsis, the enzymes acyl-CoA oxidase, nucleoside hydrolase 3, oxophytodienoate reductase 3 (OPR3), alkene oxide synthase (AOS), and dihydroflavonol 4-reductase all contain G-box elements in their promoter regions and are induced by JA (https://www.arabidopsis.org/).” in the paragraph 1 of section 2 (L91-L94).

Reference:

Ko, D. K., and Brandizzi, F. (2022). Author Correction: Transcriptional competition shapes proteotoxic ER stress resolution (Nature Plants, (2022), 8, 5, (481-490), 10.1038/s41477-022-01150-w). Nat. Plants 8, 1202. doi: 10.1038/s41477-022-01265-0.)

Suekawa, M., Fujikawa, Y., and Esaka, M. (2018). Two G-box-like elements essential to high gene expression of SlAKR4B in tomato leaves. Biosci. Biotechnol. Biochem. 82, 425–432. doi: 10.1080/09168451.2018.1429887.

Comment 7: Line 89, what chromatin-modifying enzymes are regulated by MYC?

Response 7: Thank you for your comment. We have added “MYC2 can activate downstream response genes by modifying histones. For example, it can affect histone H3 methylation of the salt stress response gene [27]. MYC2 can also form a complex with the MED25 subunit of the medium complex, thus recruiting the histone acetylase HAC1, histone acetyltransferase GCN5, and nucleosome remodeling protein SPLAYED to the promoter of downstream genes and thereby selectively regulating the state of target histones [28,29].” in the paragraph 1 of section 2 (L108-L113).

Comment 8: Line 104, what photoreceptors could be involved in the blue light sensing and MYC2 pathway? Is this phenomenon coordinated with photoperiod or circadian responses?

Response 8: Thank you for your comment. We have added “In addition to the regulation of MYC2 by JAZ, photoreceptors could be involved in blue-light sensing and the MYC2 pathway, and this phenomenon coordinated with photoperiod or circadian responses. For example, a blue light signal can induce the expression of MYC2/MYC4 through CRYPTOCHROME1 (CRY1) signal transduction to activate NST1, further activate the transcription network of secondary cell wall (SCW), and regulate the thickening of SCW in fibroblasts [40]. MYC2/ZBF1 can also mediate the interaction between light signals and JA signals, participate in the regulation of the cryptochrome-mediated blue light signal pathway, and play a role in blue light-mediated photomorphological growth. When plant seedlings are exposed to light, the blue light receptor CRYs sense the light signal and induce the biosynthesis of JA, thus activating the transcription factor MYC2/3/4, further activating the expression of the light morphogenesis regulatory gene HY5. In addition, cry1 and cry2 photoreceptors can activate the negative blue light response regulator MYC2, which can directly blind to the G-box in the plant pigment SPA1 promoter to regulate its expression [41]. SPA1 is a protein involved in regulating circadian rhythm and light signal transmission. MYC2 and SPA1 can inhibit light morphological growth, negatively regulate blue light-mediated light morphological growth, and inhibit blue light-regulated gene expression in the dark [42]. The circadian response enables plants to best respond to environmental challenges. The biosynthesis of JA is controlled by the biological clock, and its accumulation follows the law of rising in the day and falling at night. Therefore, many transcription factors, including MYC2, regulating JA signaling are also controlled by the biological clock and are governed by similar laws [43].” in the paragraph 3 of section 2 (L147-167).

Comment 9: In section 3, What are the most synthesized terpenes with repellent/insecticide properties in Arabidopsis?

Response 9: Thank you for your comment. Linalool (3,7-dimethyl-1,6-octadien-3-ol) is the more commonly synthesized terpene. Recent studies have revealed a complex interplay between pollinator attraction and plant defense mediated by linalool and its derivatives, from the smallest (Arabidopsis, Mitella) to the largest (Datura) flowers. And fig wasps, fungus gnats, and moths of all sizes show remarkable electrophysiological, neural, and behavioral sensitivity to different enantiomers and quantitative ratios of linalool in floral bouquets (Raguso, 2016).

Reference

Raguso, R. A. (2016). More lessons from linalool: Insights gained from a ubiquitous floral volatile. Curr. Opin. Plant Biol. 32, 31–36. doi: 10.1016/j.pbi.2016.05.007.

Comment 10: What are the most abundant flavonoids with insecticide/repellent properties in Helicoverpa?

Response 10: Thank you for your comment. Flavonoids are important plant secondary metabolites that protect plants from herbivorous stress. Plants have different modes of action for different insects, bollworms are chewing insects, and the most common compounds identified in peanuts eroded by bollworms are chlorogenic, syringic, quercetin (flavonoids), and ferulic acids. The number of these 4 compounds varies by genotype and therapeutic effect (War et al., 2016).

Reference

War, A. R., Sharma, S. P., and Sharma, H. C. (2016). Differential Induction of Flavonoids in Groundnut in Response to Helicoverpa armigera and Aphis craccivora Infestation. Int. J. Insect Sci. 8, IJIS.S39619. doi: 10.4137/ijis.s39619.

Comment 11: Are PR proteins also regulated by MYC2 for positive interactions against fungi or insects?

Response 11: Thank you for your comment. The defense response of plants is regulated by a highly interconnected signal network, in which the plant hormone JA plays a central role. JA has been proved to up-regulate the expression of many PR genes. MYC2, as the core transcription factor of JA signal pathway, also participates in the activation of PR gene. Overexpression of OsMYC2 results in the up-regulation of PR gene and bacterial blight resistance in rice (Uji et al., 2016). The binding site of MYC2 transcription factor was found in the promoter of two PR5 (Pp-TLP1 and Pp-TLP2) genes isolated from peach and PR5 (PdPR5-1) gene isolated from plum (El-kereamy et al., 2011; Sherif et al., 2012).

We have added “Overexpression of OsMYC2 induces up-regulation of PR gene and bacterial blight resistance in rice [66].” in the paragraph 1 of section 3 (L219-L220).

Reference:

Uji, Y., Taniguchi, S., Tamaoki, D., Shishido, H., Akimitsu, K., and Gomi, K. (2016). Overexpression of OsMYC2 results in the up-regulation of early JA-rresponsive genes and bacterial blight resistance in rice. Plant Cell Physiol. 57, 1814–1827. doi: 10.1093/pcp/pcw101.

El-kereamy, A., El-sharkawy, I., Ramamoorthy, R., Taheri, A., Errampalli, D., Kumar, P., et al. (2011). Prunus domestica pathogenesis-related protein-5 activates the defense response pathway and enhances the resistance to fungal infection. PLoS One 6, 1–11. doi: 10.1371/journal.pone.0017973.

Sherif, S., Paliyath, G., and Jayasankar, S. (2012). Molecular characterization of peach PR genes and their induction kinetics in response to bacterial infection and signaling molecules. Plant Cell Rep. 31, 697–711. doi: 10.1007/s00299-011-1188-6.

Comment 12: Drought and freezing stress response both shares the osmolyte production response for liquid water retention.

Response 12: Thank you for your comment. Studies have shown that plants can regulate their resistance to drought and cold by regulating the content of osmotic solute (Du et al., 2021; Twaij and Hasan, 2022). Interestingly, the IIIe bHLHs and the IIIb bHLH transcription factors bHLH3, bHLH13, bHLH14, and bHLH17 have antagonistic roles in regulating physiological processes such as chlorophyll accumulation, photochemical efficiency of PHOTOSYSTEM II (PSII), and membrane ion permeation (Qi et al., 2015b).

We have added “At the same time, JA can also induce SN13 expression in rice, thereby improving rice resistance to drought, cold, and freezing by affecting rice membrane integrity and osmotic matter accumulation [79].” in the paragraph 2 of section 3 (L250-L252).

Reference:

Du, B., Chen, N., Song, L., Wang, D., Cai, H., Yao, L., et al. (2021). Alfalfa (Medicago sativa L.) MsCML46 gene encoding calmodulin-like protein confers tolerance to abiotic stress in tobacco. Plant Cell Rep. 40, 1907–1922. doi: 10.1007/s00299-021-02757-7.

Twaij, B. M., and Hasan, M. N. (2022). Bioactive Secondary Metabolites from Plant Sources: Types, Synthesis, and Their Therapeutic Uses. Int. J. Plant Biol. 13, 4–14. doi: 10.3390/ijpb13010003.

Qi, T., Wang, J., Huang, H., Liu, B., Gao, H., Liu, Y., et al. (2015b). Regulation of jasmonate-induced leaf senescence by antagonism between bHLH subgroup IIIe and IIId factors in Arabidopsis. Plant Cell 27, 1634–1649. doi: 10.1105/tpc.15.00110.

Comment 13: In section 4, does MYC regulation of only nuclear encoded genes? Can MYC regulate plastidial genes?

Response 13: Thank you for your question. Under normal circumstances, we believe that there is no free gene in the cytoplasm, and the gene is either in the nucleus or in the organelle. Therefore, we believe that MYC and other transcription factors can only regulate the genes encoded in the nucleus. However, if there are genes in the cytoplasm, such as plasmid genes introduced into the cytoplasm by transfection, MYC can also regulate their expression.

Comment 14: In senescence, does MYC2 signaling activates the expression of transporters, ion pumps, degradation enzymes, cell wall remodeling enzymes?

Response 14: Thank you for your comment. We have added “During the senescence of rape leaves, MYC can be induced and activated by ABA, and then up-regulate the expression of AMY3 and BAM1, which are related to starch degradation, and the sucrose transporters SUT1, SUT4 and SWEET11 [85]. In the process of Solanum lycopersicum leaf senescence, SlMYC2 can enhance the expression of SIPAO, which encodes a chlorophyll degrading enzyme and plays an active role in Solanum lycopersicum leaf senescence [86].” in the paragraph 1 of section 4 (L278-283). We can't find references about MYC2 signaling activating the expression of ion pumps and cell wall remodeling enzymes in senescence.

Comment 15: Paragraph at line 188. Some root remodeling response involve JA and brassinosteroids for nitrogen and phosphate uptake.

Response 15: Thank you for your comment. We have added “Plant roots have evolved different systems for nitrogen uptake. Auxin, CTK, ABA, ETH, GA, BR, and JA can regulate NO3- and NH4+ uptake by regulating the transcript levels or transport activities of the NPF/NRT1, NRT2, and AMTs families in various plants [90]. In addition, JA can also mediate the expression of OsWRKY28 to enhance rice root elongation and phosphate absorption [91]. Simultaneously, JA can also enhance the nitrogen absorption of legumes by enhancing the formation of root nodules, thus promoting the growth of roots [92].” in paragraph 2 of section 4 (L300-307).

Comment 16: Paragraph at line 202. Are there intertwined interactions between MYC2 with gibberellin-response genes for fruit maintenance and seed development? Or interactions with ethylene and ABA-response genes for fruit ripening?

Response 16: Thank you for your comment. We have added “ABA induces the expression of MYC2 and MYB1R1 and activates the PbFAD3a promoter, contributing to the formation of russet pear skin [96].” in paragraph 3 of section 4 (L318-319). We have added “JA and ET participate in the gibberellin-induced ovule programmed cell death process in seedless pear '1913' (pyrus hybrid) [97].” in paragraph 3 of section 4 (L325-326).

In the manuscript (L308-314) we show that JA and MYC2 promote the expression of 1-aminocyclopropane-1-carboxy acid oxidase and synthetase and ET synthesis to promote fruit ripening. MYC2 can not only directly promote the expression of 1-aminocyclopropane-1-carboxylate oxidase and synthase, but also promote the expression of 1-aminocyclopropane-1-carboxylate synthase through the ERF3 pathway. MYC2 can interact with ERF2, thereby releasing the binding between ERF2 and ERF3 and enhancing the transcriptional activation of ACS1 by ERF3.

Comment 17: MYC2-regulated morphogenesis of flower androcium occurs in dioecious/monoecious plants? Or hermaphrodite flowers? Or occurs for self-compatibility/incompatibility pollination?

Response 17: Thank you for your comment. MYC2-regulated morphogenesis of flower androecium occurs in monoecious plants, such as Chestnut (Castanea mollisima) (Cheng et al., 2022).

Reference:

Cheng, H., Zha, S., Luo, Y., Li, L., Wang, S., Wu, S., et al. (2022). JAZ1‐3 and MYC2‐1 Synergistically Regulate the Transformation from Completely Mixed Flower Buds to Female Flower Buds in Castanea mollisima. Int. J. Mol. Sci. 23. doi: 10.3390/ijms23126452.

Comment 18: In section 5, regarding bionsynthesis genes for paclitaxel production, are these mostly regulated by MYC2? Or are only branches of the biosynthesis pathway regulated by MYC2?

Response 18: Thank you for your comment. Jasmonates (JAs) is the most effective inducer of biosynthesis of various secondary metabolites, which can greatly promote the biosynthesis of taxol. However, different from most other secondary metabolites, taxol has a highly complicated chemical structure and needs more than 19 enzymes to catalyze the diterpene universal precursor, geranylgeranyl pyrophosphate, to the final product. Among 14 known enzymes, taxane synthase (TASY) and 10-deacetylbaccatin III-10β-O-acetyltransferase (DBAT) were functioned crucially for taxol biosynthesis; TASY decides the flux because it is the first enzyme catalyzing universal diterpene precursor and produces crucial skeleton; DBAT acts as a rate-limiting enzyme(Zhang et al., 2015; Li et al., 2017) Additionally, TASY and DBAT are early JA-responsive genes (Zhang et al., 2018). Currently, jasmonate ZIM domain (JAZ) and its interactors, such as MYC2, constitute the main JA signal transduction cascade, and such a cascade fails to directly regulate all the taxol biosynthesis genes (Chen et al., 2022) Only TASY is clearly defined as the JA response mechanism, TcMYC2a can regulate taxol biosynthesis directly or through ERF regulator (Li et al., 2017; Zhang et al., 2017, 2018).There are 12, 10 and 11 paclitaxel biosynthesis genes promoters that could be activated by MYC2, MYC3 and MYC4 (Cui et al., 2019).

Reference

Li, B. J., Wang, H., Gong, T., Chen, J. J., Chen, T. J., Yang, J. L., et al. (2017). Improving 10-deacetylbaccatin III-10-β-O-acetyltransferase catalytic fitness for Taxol production. Nat. Commun. 8, 1–13. doi: 10.1038/ncomms15544.

Zhang, M., Li, S., Nie, L., Chen, Q., Xu, X., Yu, L., et al. (2015). Two jasmonate-responsive factors, TcERF12 and TcERF15, respectively act as repressor and activator of tasy gene of taxol biosynthesis in Taxus chinensis. Plant Mol. Biol. 89, 463–473. doi: 10.1007/s11103-015-0382-2.

Zhang, M., Jin, X., Chen, Y., Wei, M., Liao, W., Zhao, S., et al. (2018). TcMYC2a, a basic helix–loop–helix transcription factor, transduces JA-signals and regulates taxol biosynthesis in taxus chinensis. Front. Plant Sci. 9, 1–13. doi: 10.3389/fpls.2018.00863.

Chen, L., Wu, L., Yang, L., Yu, H., Huang, P., Wang, Y., et al. (2022). TcJAV3–TcWRKY26 Cascade Is a Missing Link in the Jasmonate-Activated Expression of Taxol Biosynthesis Gene DBAT in Taxus chinensis. Int. J. Mol. Sci. 23, 13194. doi: 10.3390/ijms232113194.

Zhang, M., Chen, Y., Nie, L., Jin, X., Fu, C., and Yu, L. (2017). Molecular, structural, and phylogenetic analyses of Taxus chinensis JAZs. Gene 620, 66–74. doi: 10.1016/j.gene.2017.04.005.

Cui, Y., Mao, R., Chen, J., and Guo, Z. (2019). Regulation mechanism of MYC family transcription factors in jasmonic acid signalling pathway on taxol biosynthesis. Int. J. Mol. Sci. 20. doi: 10.3390/ijms20081843.

Comment 19: It seems MYC2 frequently regulates terpenoids biosynthesis pathways in all given examples.

Response 19: Thank you for your comment. Star proteins such as paclitaxel, artemisinin and tanshinone are terpenoids, so there are many descriptions about them in this paper. However, MYC transcription factors are involved in the regulation of secondary metabolites such as alkaloids, flavonoids and thioglycosides in addition to terpenes. Many genes that are regulating flavonoid biosynthesis, such as MYB/PAP and EGL, are also regulated by MYC2 transcription factor, and in tobacco, MYC2 also regulates the expression of key genes for nicotine biosynthesis (Afrin et al., 2015).

Reference:

Afrin, S., Huang, J. J., and Luo, Z. Y. (2015). JA-mediated transcriptional regulation of secondary metabolism in medicinal plants. Sci. Bull. 60, 1062–1072. doi: 10.1007/s11434-015-0813-0.

Comment 20: Are there MYC2 regulated terpenoid biosynthesis for eucalyptus, pine or conifers?

Response 20: Thank you for your comment. Eucalypts are well known for their essential oils, which are typically stored in the sub-dermal secretory cavities of mature leaves. These oils are a complex mixture of monoterpenes, sesquiterpenes and FPCs, which is reviewed in (Keszei et al., 2008). Synthesis of certain A. thaliana sesquiterpenes, including β-caryophyllene, is induced by both GA and JA and involves the TF MYC2, which is repressed by DELLA proteins. This indicates that the production of these sesquiterpenes is integrated with the GA and JA signaling pathways (Hong et al., 2012). So we speculate that MYC2 may also be involved in the synthesis of terpenoids in Eucalyptus, but the specific mechanism has not yet been resolved.

Reference:

Keszei, A., Brubaker, C. L., and Foley, W. J. (2008). A molecular perspective on terpene variation in Australian Myrtaceae. Aust. J. Bot. 56, 197–213. doi: 10.1071/BT07146.

Hong, G. J., Xue, X. Y., Mao, Y. B., Wang, L. J., and Chen, X. Y. (2012). Arabidopsis MYC2 interacts with DELLA proteins in regulating sesquiterpene synthase gene expression. Plant Cell 24, 2635–2648. doi: 10.1105/tpc.112.098749.

Comment 21: In section 6, What characteristics should accomplish chassis cells for plant metabolite production? What are their advantages and limitations?

Response 21: Thank you for your comment. We have added “The chassis cells produced using plant metabolites include plant chassis and microbial cell chassis. Microbial cell chassis development utilizes E. coli and S. cerevisiae as typical hosts. E. coli is the most widely prokaryotic host used to produce heterogenous metabolites. As chassis cell, it has the characteristics of rapid/easy growth, high product yield, and high cost efficiency. Further, the availability of various expression vectors and strains, the operation technology and the relative ease of product purification make them attractive host for industrial applications. However, the lack of cell intima as well as post-translational modification limit its use as a selection chassis for many plant natural products [140]. in paragraph 4 of section 6 (L479-487).

Comment 22: How many genes can be incorporated into an artificial chromosome?

Response 22: Thank you for your comment. The field of synthetic biology is advancing rapidly, and while it is now possible to assemble complete genomes and use them to reboot cells, designing gene networks that encode new biological functions remains very challenging, even on a small scale. In fact, most previous genetic engineering efforts have addressed fewer than 10 genes. Currently, the most reported vector has a size of 50kb and 17 genes, and the Golden Gate cloning and MoClo systems are used to quickly assemble a multigene structure (Werner et al., 2012). And our laboratory is currently constructing a JA-inducible promoter switched by the MYC2 position, which has reached 16 genes.

Reference:

Werner, S., Engler, C., Weber, E., Gruetzner, R., and Marillonnet, S. (2012). and the MoClo system. Structure 3, 1–6.

Comment 23: How efficient are the post-translational modifications in Saccharomices gene expression systems?

Response 23: Thank you for your comment. In eukaryotes, frequent PTMs include phosphorylation, acetylation, methylation, glycosylation, lipidation, flavinylation, ubiquitylation, and proteolysis. These general terms can comprise a variety of more specific chemical modifications, even within the same organism. Protein phosphorylation is the best studied and arguably the most frequent PTM in Saccharomyces cerevisiae. Protein kinases are typically rather specific, while the about 40 protein phosphatases in yeast exhibit somewhat lower specificity for their targets. About half of the metabolic enzymes in yeast can be regulated by at least two different processes, at the gene expression level and post-translational phosphorylation, but more experiments are needed to verify post-translational modification efficiency (Oliveira and Sauer, 2012).

Reference:

Oliveira, A. P., and Sauer, U. (2012). The importance of post-translational modifications in regulating Saccharomyces cerevisiae metabolism. FEMS Yeast Res. 12, 104–117. doi: 10.1111/j.1567-1364.2011.00765.x.

Comment 24: What molecules have been successfully produced using microalgae expression systems?

Response 24: Thank you for your comment. The ability of algae to accumulate metals and reduce metal ions makes them a strong competitor for the biosynthesis of nanoparticles, so they are called biological nanofactories, because both living and dead dry biomass are used to synthesize metal nanoparticles. Microalgae are an important part of the biodiversity of the earth. They are usually single-celled colony forming or filamentous photosynthetic microorganisms, including several legal branches such as Chlorophyta, Charophyta and Diatoma. The research shows that the whole cell of pineapple algae (filamentous cyanobacteria) can effectively promote the production of gold, silver and platinum nanoparticles. Biosynthesis of cash, silver and platinum nanoparticles by cyanobacteria strains of Anabaena flos aquatica and Calophyllum pulvinate. Once synthesized in the cell, nanoparticles are released into the culture medium, where they form stable colloid for easy recovery. It is reported that Lyngbya Majuscule and Chlorella are cost-effective silver nanoparticles synthesis chassis. The dried edible algae (Spirulina) can be used for the extracellular synthesis of gold, silver and gold/silver bimetallic nanoparticles. It is also reported that the synthesis of extracellular metal biological nanoparticles from Sargassum sargassum and Alvarez kappa. Bioreduction of Au (III) - Au (0) by using the biomass of brown algae and fucoidan, and biosynthesis of Au nanoparticles by using the biomass of red algae (Chondrus crispus) and green algae (Spyrogira insignis). Algae treatment is relatively convenient, less toxic and less harmful to the environment. The synthesis can be carried out at ambient temperature and pressure and in simple water medium with normal pH value [146–148].

We have added “and have successfully synthesized metal nanoparticles [146–148].” in the paragraph 3 of section 6 (L499-500).

Reviewer 4 Report

This review summarizes the function of the master transcription factor MYC2 in jasmonic acid signaling in plant development, growth, and production of specialized metabolites. It also discusses the potential use of regulatory machinery including MYC2 as tools for synthetic biology.
The following excellent reviews have been published related to MYC2. MYC2 functions in plant development, growth, and production of specialized metabolites are well summarized by Song et al. in 2022.

Song C, Cao Y, Dai J, Li G, Manzoor MA, Chen C, Deng H. The Multifaceted Roles of MYC2 in Plants: Toward Transcriptional Reprogramming and Stress Tolerance by Jasmonate Signaling. Front Plant Sci. 2022 Apr 25;13:868874. doi: 10.3389/fpls.2022.868874. PMID: 35548315; PMCID: PMC9082941.

Zhai Q, Deng L, Li C. Mediator subunit MED25: at the nexus of jasmonate signaling. Curr Opin Plant Biol. 2020 Oct;57:78-86. doi: 10.1016/j.pbi.2020.06.006. Epub 2020 Aug 7. PMID: 32777679.

Goossens J, Mertens J, Goossens A. Role and functioning of bHLH transcription factors in jasmonate signalling. J Exp Bot. 2017 Mar 1;68(6):1333-1347. doi: 10.1093/jxb/erw440. PMID: 27927998.

Kazan K, Manners JM. MYC2: the master in action. Mol Plant. 2013 May;6(3):686-703. doi: 10.1093/mp/sss128. Epub 2012 Nov 9. PMID: 23142764.

Unfortunately, this review does not find any advantages over the published reviews. The perspective from synthetic biology is original, but the review cannot show the published literature to show examples of the actual use of regulatory machinery including MYC2, in synthetic biology. In addition, it is unclear what makes the regulatory machinery including MYC2 a good tool in synthetic biology compared to other transcription factor-based systems.

For these reasons, we cannot recommend the publication of this review.

Author Response

Response to Reviewer 4 Comments

Dear Reviewers:

Thank you for your comments on our manuscript entitled “MYC2: A Master Switch for Plant Physiological Processes and Specialized Metabolite Synthesis” (ijms-2099327). We regret that the manuscript cannot be approved by you. Those comments were very helpful for improving our manuscript. We have answered reviewer’s comments. The main corrections in the paper and the responds to the reviewer’s comments are as follows:

Comments to the Author

This review summarizes the function of the master transcription factor MYC2 in jasmonic acid signaling in plant development, growth, and production of specialized metabolites. It also discusses the potential use of regulatory machinery including MYC2 as tools for synthetic biology.

The following excellent reviews have been published related to MYC2. MYC2 functions in plant development, growth, and production of specialized metabolites are well summarized by Song et al. in 2022.

Song C, Cao Y, Dai J, Li G, Manzoor MA, Chen C, Deng H. The Multifaceted Roles of MYC2 in Plants: Toward Transcriptional Reprogramming and Stress Tolerance by Jasmonate Signaling. Front Plant Sci. 2022 Apr 25;13:868874. doi: 10.3389/fpls.2022.868874. PMID: 35548315; PMCID: PMC9082941.

Zhai Q, Deng L, Li C. Mediator subunit MED25: at the nexus of jasmonate signaling. Curr Opin Plant Biol. 2020 Oct;57:78-86. doi: 10.1016/j.pbi.2020.06.006. Epub 2020 Aug 7. PMID: 32777679.
Goossens J, Mertens J, Goossens A. Role and functioning of bHLH transcription factors in jasmonate signalling. J Exp Bot. 2017 Mar 1;68(6):1333-1347. doi: 10.1093/jxb/erw440. PMID: 27927998.

Kazan K, Manners JM. MYC2: the master in action. Mol Plant. 2013 May;6(3):686-703. doi: 10.1093/mp/sss128. Epub 2012 Nov 9. PMID: 23142764.

Unfortunately, this review does not find any advantages over the published reviews. The perspective from synthetic biology is original, but the review cannot show the published literature to show examples of the actual use of regulatory machinery including MYC2, in synthetic biology. In addition, it is unclear what makes the regulatory machinery including MYC2 a good tool in synthetic biology compared to other transcription factor-based systems.

For these reasons, we cannot recommend the publication of this review.

Response: Thank you for your comment. Jasmonate (JA) signal pathway is a very important plant hormone in plants, and the importance of MYC2 as a core transcription factor in JA signal pathway is self-evident, so there are many articles about it. However, compared with other reviews, this review summarizes more and updated proteins that interact or regulate with MYC2 (such as SlTOR, SUMO, PPCs, SGAs and AabHLHs), which are more comprehensive and detailed in content. Moreover, the pictures drawn in this article are exquisite and the tables made are relatively complete, which makes this article competitive in the same type of review. I believe that interesting pictures and prospects have certain attraction for readers.

In addition, the complexity of the JA signal pathway mediated by MYC2 is not clear in the previous article, and the description of the regulatory impact of the signal pathway subnetworks on the wider network through MYC2 is not clear. So how to apply these subnetworks regulated by MYC2 to drive the activation of JA signals? I think combining the concept of synthetic biology will be an effective method.

Synthetic biology is an emerging field guided by engineering ideas and combined with multiple disciplines. It enables organisms or cells to have new capabilities through a series of redesign and technological transformation. It is important to build a series of new standardized biological elements, components and systems in this process, but the most important thing is how to design and conceive to achieve an ideal synthetic biological system. Therefore, we put forward the prospect of using MYC2 as a switch to control the physiological process of plants and the production of natural metabolites of plants, design and construct artificial chromosomes, add JA inducible promoters and MYC2 switches, and transfer to selected chassis plants or cells to reconstruct the synthesis process of special products to obtain high content of target products. This is a new point of view that has not been put forward in other articles, and we think it is of great significance to promote the in-depth use of MYC2.

In the application of synthetic biology, transcription factors and promoters that naturally exist in plants have been developed as molecular switches and promoters for efficient synthesis of specialized metabolites. For example, in rice, scientists have constructed a construction containing eight anthocyanidin-related genes (two regulatory genes from maize and six structural genes from Perilla frutescens), which are driven by endospermspecific promoters, to produce a new bioenhanced germplasm "purple endosperm rice" (called "Zijingmi" in Chinese) with high anthocyanin content and antioxidant activity [137]. In purple-leaf rice, the regulatory proteins include OsC1 (the homolog of the MYB-R2R3-type transcription factor Colored 1 encoded by ZmC1 in maize), OsB1, and OsB2 (the homologs of the basic helix-loop-helix [bHLH]-type transcription factor Booster 1 of maize), and OsWD40 (a WD40-type transcription factor); these proteins interact to form the MYB–bHLH–WD40 (MBW) complex, which activates the genes for anthocyanin biosynthesis in leaves (Sakamoto et al., 2001). Researcher selected maize ZmLc (Leaf color) and ZmPl (Purple leaf), which encode the bHLH-type and MYB-type transcription factors, respectively, to activate the anthocyanin biosynthesis genes; ZmPl is a homolog of ZmC1 but has stronger activity [138,139].

This success could be attributed to several factors in the design. First, researcher used both the transgenic and endogenous regulatory genes where possible. For example, the transgenic ZmLc and ZmPl and the endogenous OsWD40 rebuilt the MBW complex, which enhanced or activated the expression of some other endogenous genes for gene regulation (e.g., OsWD40) and for biosynthesis, decoration, and transport of the anthocyanin products. In fact, the two lines that lacked only ZmLc produced white grains, confirming the requirement of ZmLc for this engineered anthocyanin biosynthesis. Second, researcher supplemented endogenous genes with transgenes. For example, although three endogenous genes (OsCHS, OsCHI, and OsF3′H) were activated by the rebuilt MBW complex, the absence of the corresponding transgenes in the lines resulted in lower anthocyanin contents and lower antioxidant activity. Therefore, introduction of the two necessary regulatory genes together with the complete set of structural genes successfully produced and enhanced anthocyanin biosynthesis in the rice endosperm. This shows that our outlook may be applied to synthetic biology, and it is very important.

MYC2, as a star protein, not only participates in the regulation of plant growth and development through JA signal pathway, but also mediates plant defense response to various biological and abiotic stresses through the interaction with ABA, ET, GA and SA. The upstream of MYC2 is regulated by COI1, NINJA, TPL, and JAZs, while the downstream can affect many transcription factors such as TPs, bHLH series, and ERF series. The identity of MYC2 protein as the transduction center of a variety of key signals is the reason for deciding to use MYC2 as the switch to build the secondary metabolite synthesis system. Many articles with MYC2 as the main body will be published every year, which can also explain the importance of MYC2 laterally. Thank you for your understanding.

Based on the above, we have added “In the application of synthetic biology, transcription factors and promoters that naturally exist in plants have been developed as molecular switches and promoters for efficient synthesis of specialized metabolites. For example, in rice, scientists have constructed a construction containing eight anthocyanidin-related genes (two regulatory genes from maize and six structural genes from Perilla frutescens), which are driven by endospermspecific promoters, to produce a new bioenhanced germplasm "purple endosperm rice" (called "Zijingmi" in Chinese) with high anthocyanin content and antioxidant activity [137]. Researcher selected maize ZmLc (Leaf color) and ZmPl (Purple leaf), which encode the bHLH-type and MYB-type transcription factors, respectively, to activate the anthocyanin biosynthesis genes [138,139].” in the paragraph 3 of section 6 (L467-L476).

Reference:

  1. Zhu, Q.; Yu, S.; Zeng, D.; Liu, H.; Wang, H.; Yang, Z.; Xie, X.; Shen, R.; Tan, J.; Li, H.; et al. Development of “Purple Endosperm Rice” by Engineering Anthocyanin Biosynthesis in the Endosperm with a High-Efficiency Transgene Stacking System. Mol. Plant 2017, 10, 918–929, doi:10.1016/j.molp.2017.05.008.
  2. Han, Y.J.; Kim, Y.M.; Lee, J.Y.; Kim, S.J.; Cho, K.C.; Chandrasekhar, T.; Song, P.S.; Woo, Y.M.; Kim, J. Il Production of Purple-Colored Creeping Bentgrass Using Maize Transcription Factor Genes Pl and Lc through Agrobacterium-Mediated Transformation. Plant Cell Rep. 2009, 28, 397–406, doi:10.1007/s00299-008-0648-0.
  3. Bovy, A.; De Vos, R.; Kemper, M.; Schijlen, E.; Almenar Pertejo, M.; Muir, S.; Collins, G.; Robinson, S.; Verhoeyen, M.; Hughes, S.; et al. High-Flavonol Tomatoes Resulting from the Heterologous Expression of the Maize Transcription Factor Genes LC and C1. Plant Cell 2002, 14, 2509–2526, doi:10.1105/tpc.004218.
  4. Sakamoto, T. Ohmori, K. Kageyama. The Purple leaf (Pl) locus of rice: the Plw allele has a complex organization and includes two genes encoding basic helix-loop-helix proteins involved in anthocyanin biosynthesis. Plant Cell Physiol., 42 (2001), pp. 982-991.

Round 2

Reviewer 4 Report

The description of the physiological functions of MYC2 in the revised manuscript has been improved by adding information not included in Song et al., 2022, such as interactions with other plant hormones, according to the other reviewer's suggestion. However, there is still a problem in the description of the prospect of using MYC2 in synthetic biology. The reviewer agrees with the advantage of using MYC2 in synthetic biology using plants as hosts, as explained in the description added in the revised manuscript and the author's response. However, it is still unclear what is the advantage of using MYC2 in synthetic biology using microorganisms as hosts.

 1) Many other plant transcription factors and their cis-elements were used in yeast for synthetic biology tools (Naseri et al., 2017). The advantage of using MYC2 should be clearly stated based on the evidence, literature, or working model, compared with other transcription factors.

Naseri, G. et al. (2017). Plant-Derived Transcription Factors for Orthologous Regulation of Gene Expression in the Yeast Saccharomyces cerevisiae. ACS Synth Biol. 6, 1742-175

2) The manuscript described that specialized metabolite production could be induced by exogenous JA (Line 481). However, the auxin degron system (Nishimura et al., 2009), a synthetic biology tool using the other plant hormone auxin, is used in various organisms. The auxin degron system was originally a method to control protein degradation. However, it is also used for artificial auxin-responsive promoters in combination with CRISPR transcription factors not only in the plant but also in yeast (Khakhar et al., 2016,  2018). What is the advantage of using jasmonic acid as an inducer instead of auxin?

Nishimura, K., Fukagawa, T., Takisawa, H. et al. An auxin-based degron system for the rapid depletion of proteins in nonplant cells. Nat Methods 6, 917–922 (2009). https://doi.org/10.1038/nmeth.1401

 Arjun Khakhar, Nicholas J. Bolten, Jennifer Nemhauser, and Eric Klavins. Cell–Cell Communication in Yeast Using Auxin Biosynthesis and Auxin Responsive CRISPR Transcription Factors. ACS Synthetic Biology 2016 5 (4), 279-286. DOI: 10.1021/acssynbio.5b00064

 Khakhar A, Leydon AR, Lemmex AC, Klavins E, Nemhauser JL. Synthetic hormone-responsive transcription factors can monitor and re-program plant development. Elife. 2018 May 1;7:e34702. doi: 10.7554/eLife.34702. PMID: 29714687; PMCID: PMC5976440.

3) Please consider mentioning the effect of endogenous jasmonic acid produced by the host organism on regulation by MYC2. Several microalgae biosynthesize jasmonic acid and conserve jasmonic acid signaling factors (Han 2017). Jasmonic acid has also been detected in E. coli and yeast, albeit in older literature (Abdala et al., 1999).

G. Abdala, O. Miersch, N. Correa & S. Rosas (1999) Detection of Jasmonic Acid in Cultures of Escherichia Coli and Saccharomyces Cerevisiae, Natural Product Letters, 14:1, 55-63, DOI: 10.1080/10575639908045435

 Guan-Zhu Han, Evolution of jasmonate biosynthesis and signaling mechanisms, Journal of Experimental Botany, Volume 68, Issue 6, 1 March 2017, Pages 1323–1331, https://doi.org/10.1093/jxb/erw470

4) In Figures 2a-c, the scientific names are used, but in Figures 2d-e, the metabolite names are used. The names listed should be consistent.

Author Response

Response to Reviewer 4 Comments

Dear Reviewer:

Thank you for your comments on our manuscript entitled “MYC2: A Master Switch for Plant Physiological Processes and Specialized Metabolite Synthesis” (ijms-2099327). Those comments were very helpful for improving our manuscript. We believed we have answered reviewer’s all comments. The main corrections in the paper and the responds to the reviewer’s comments are as follows:

Comments and Suggestions for Authors

The description of the physiological functions of MYC2 in the revised manuscript has been improved by adding information not included in Song et al., 2022, such as interactions with other plant hormones, according to the other reviewer's suggestion. However, there is still a problem in the description of the prospect of using MYC2 in synthetic biology. The reviewer agrees with the advantage of using MYC2 in synthetic biology using plants as hosts, as explained in the description added in the revised manuscript and the author's response. However, it is still unclear what is the advantage of using MYC2 in synthetic biology using microorganisms as hosts.

Response: Thank you for your comment. Transcriptional regulation plays an important role in gene expression (Holtz and Keasling, 2010), therefore, it is crucial to select an appropriate transcription factor to establish an orthogonal regulatory system for synthetic biology applications (Purcell et al., 2014). Plants are cost-effective bioreactors that require sunlight, minerals, water and air to produce food, feed and fiber. Bioengineering of plants requires regulatory switches to control functional modules. Plant hormones and their associated signaling pathways are examples of such switches by mediating signal-specific responses with high sensitivity and efficiency. Reasonable reconstruction of metabolic pathways and regulatory systems is one of the important goals of synthetic biology and metabolic engineering. There are good examples of successful reprogramming of cellular functions by utilizing different biosensors, genetic circuits, and engineering pathways (Niu et al., 2018).

In contrast to constitutive gene expression systems, precise regulation of inducible and repressible gene expression systems will facilitate dynamic gene regulation for optimal production of valuable chemicals while avoiding burden effects that negatively affect host cell growth. An ideal control system should allow rapid and precise regulation of a target gene between the “ON” and “OFF” states or even simultaneous switching of different genes to the ON or OFF state (Gossen and Bujardt, 1992; Zhou and Zeng, 2015; Urlinger, 2000). The researcher report the development of a malO-based genetic toolbox derived from the operator box promoter in malA, enabling gene regulation via compatible “ON” and “OFF” switches. This regulatory system provides a comprehensive genetic toolbox for controlling the expression of genes in biosynthetic pathways and regulatory networks to optimize the production of valuable chemicals and proteins in Bacillus subtilis (Fu et al., 2022). Compatible “ON” and “OFF” switch functions for controlling the expression of genes in biosynthetic pathways and regulatory networks can also be achieved through rational design of MYC2-based switches. The JA pathway regulates the synthesis of multiple secondary metabolites. Therefore, it is more helpful to efficiently induce the expression of pharmaceutical products by heterologous reconstruction of JA regulation in microorganisms and establishment of a system that uses JA-JAZ-MYC2 as a switch to induce the expression of jasmonate inducible promoters.

The JA core signaling module includes F-box protein SCFCOI1, JAZ protein and transcription factor MYC2. In the JA signaling pathway, MYC is the main downstream effector transcription factor, while JAZ is the negative regulatory protein of JA signaling, which can bind and inhibit the transcription factor activity of MYC2. In the absence of JA, JAZ interacts with MYC2 and recruits TPL through NINJA, inhibiting MYC2 from activating the expression of JA-responsive genes. When JA exists as an activation signal, JA-Ile promotes the formation of the SCFCOI1-JAZ co-receptor complex, degrades JAZ through 26S proteasome ubiquitination, and releases the inhibition of JAZ on MYC2. On the other hand, MYC2 can activate JA-induced bHLH protein and negatively regulate the JA signaling pathway by impairing the formation of MYC2-MED25 complex, leading to the termination of JA signaling.

Nearly half of the more than 2,000 plant transcription factors are considered plant-specific, and for the regulation of complex gene expression systems in synthetic biology projects, TFs with different transcriptional activation capabilities are often required. Due to the differences between prokaryotic systems and eukaryotic systems, it may be more advantageous to introduce a de novo synthesis system in a relatively “pure” prokaryotic system. Although there are no direct literature reports on the actual use of MYC2 in microbial hosts, it has been reported that E. coli strains can synthesize jasmonic acid, and jasmonic acid has also been identified as an endogenous compound used in yeast (Saccharomyces cerevisiae) (Abdala et al., 1999). Therefore, MYC2 has the potential to regulate the jasmonic acid signaling pathway in microorganisms.

Comment 1: Many other plant transcription factors and their cis-elements were used in yeast for synthetic biology tools (Naseri et al., 2017). The advantage of using MYC2 should be clearly stated based on the evidence, literature, or working model, compared with other transcription factors.

Naseri, G. et al. (2017). Plant-Derived Transcription Factors for Orthologous Regulation of Gene Expression in the Yeast Saccharomyces cerevisiae. ACS Synth Biol. 6, 1742-175

Response 1: Thank you for your comment. JA plays an important role in the growth and development of plants and the defense against the outside world. The MYC2 transcription factor is the master regulator in the JA signaling pathway, not only controlling the biosynthesis of many important metabolites (from natural products such as nicotine and steroidal glycoalkaloids to sesquiterpene lactones artemisinin; drugs such as diterpenoid paclitaxel and monoterpene indole alkaloids, vincristine and vinblastine ) (Scossa et al., 2018; Shoji and Yuan, 2021), but also modulate JA signaling with light and phytochrome signaling and circadian rhythms, and crosstalk between JA and multiple other hormone signaling pathways (such as abscisic acid, salicylic acid, gibberellins, and auxin) (Kazan and Manners, 2013). Although other transcription factors can also respond to JA signaling, most of them are antagonistic to the MYC2 transcription factor. For example, in the response of JA-induced defense against herbivores, there are two branches of MYC2 and AP2/ERF. The ERF branched is more likely to attract herbivores to feed, while the MYC2 branch can inhibit the ERF branch and reduce the attractiveness of plants to attackers (Verhage et al., 2011). This kind of response is very important for plants, and MYC2 can regulate a variety of JA responses, which makes MYC2 more important than other transcription factors.

Through sequencing technology, it is found that the target genes directly bound by MYC2 are rich in transcription factor genes, which indicates that MYC2 is a high-level transcriptional regulatory element in the JA signaling pathway (Du et al., 2017), and the secondary transcription factors that MYC2 directly binds form A series of transcriptional cascade regulatory modules that activate and amplify the responses elicited by JA. Among these secondary transcription factors, there are many other transcription factor families, such as the NAC transcription factor family. Coronatine bacteria can directly activate MYC2 to activate three homologous NAC transcription factor genes, which inhibit the accumulation of SA, allowing the pathogen to infect plants and reproduce (Zheng et al., 2012). It can be seen that by regulating a single MYC2 transcription factor, the effect of regulating multiple transcription factors can be achieved, which can be described as twice the result with half the effort. Unlike plant-specific transcription factor families such as NAC and WRKY, MYC genes (c-Myc, MYCN, MYCL) also exist in animals, and are highly homologous to the BHLH region of MYC2 in plants (the key region for binding downstream target genes) (Dhanasekaran et al., 2022). So in research experiments using animal cells as the chassis cells, such as expressing proteins in animal cells, the MYC2 transcription factor has greater advantages in terms of stability and expression ability.

In general, transgene expression only needs to occur at a certain point or time period in order to minimize the metabolic burden on the host cell, or to control the timing of gene expression. For this reason, unlike constitutive promoters such as CaMV35S, Nos, and Ocs, which are always active, the inducible gene expression system constructed by MYC2 can more dynamically and precisely regulate gene expression for optimal production of valuable chemicals (Heiss et al., 2016). In addition, MYC2 can also bind to the promoters of key synthetic genes of natural products such as paclitaxel and artemisinin to enhance their expression (Cui et al., 2019). All of which indicate that MYC2 is more suitable as a plant transcription factor "Switches" that induce the expression of multiple downstream genesthan other plant transcription factors.

An ideal control system should allow rapid and precise regulation of a target gene between the “ON” and “OFF” states or even simultaneous switching of different genes to the ON or OFF state (Gossen and Bujardt, 1992; Zhou and Zeng, 2015; Yan, 2017). Compatible “ON” and “OFF” switch functions for controlling the expression of genes in biosynthetic pathways and regulatory networks can also be achieved through rational design of MYC2-based switches. The JA pathway regulates the synthesis of various secondary metabolites. Therefore, heterologous reconstruction of JA regulation in microorganisms and the establishment of a system that uses JA-JAZ-MYC2 as a switch to induce the expression of JA-inducible promoters are more conducive to the efficient induction of the expression of medicinal product.

We have added “The transcription factor MYC2 is a high-level transcriptional regulatory element in the JA signal pathway. MYC2 and its direct binding secondary transcription factors form a series of transcriptional cascade regulatory modules, which activate and amplify the response caused by JA, and can achieve the effect of regulating multiple transcription factors [20].” in the paragraph 3 of section 1 (L72-L76).

We have added “Transgene expression only needs to occur at a certain point or time period in order to minimize the metabolic burden on the host cell, or to control the timing of gene ex-pression. For this reason, unlike constitutive promoters such as CaMV35S, Nos, and Ocs, which are always active, the inducible gene expression system constructed by MYC2 can more dynamically and precisely regulate gene expression for optimal production of valuable chemicals [152].An ideal control system should allow rapid and precise regu-lation of a target gene between the “ON” and “OFF” states or even simultaneous switching of different genes to the ON or OFF state [153–155].Compatible “ON” and “OFF” switch functions for controlling the expression of genes in biosynthetic pathways and regulatory networks can also be achieved through rational design of MYC2-based switches.” in the paragraph 5 of section 6 (L521-L530).

Comment 2: The manuscript described that specialized metabolite production could be induced by exogenous JA (Line 481). However, the auxin degron system (Nishimura et al., 2009), a synthetic biology tool using the other plant hormone auxin, is used in various organisms. The auxin degron system was originally a method to control protein degradation. However, it is also used for artificial auxin-responsive promoters in combination with CRISPR transcription factors not only in the plant but also in yeast (Khakhar et al., 2016,  2018). What is the advantage of using jasmonic acid as an inducer instead of auxin?

Nishimura, K., Fukagawa, T., Takisawa, H. et al. An auxin-based degron system for the rapid depletion of proteins in nonplant cells. Nat Methods 6, 917–922 (2009). https://doi.org/10.1038/nmeth.1401

 Arjun Khakhar, Nicholas J. Bolten, Jennifer Nemhauser, and Eric Klavins. Cell–Cell Communication in Yeast Using Auxin Biosynthesis and Auxin Responsive CRISPR Transcription Factors. ACS Synthetic Biology 2016 5 (4), 279-286. DOI: 10.1021/acssynbio.5b00064

 Khakhar A, Leydon AR, Lemmex AC, Klavins E, Nemhauser JL. Synthetic hormone-responsive transcription factors can monitor and re-program plant development. Elife. 2018 May 1;7:e34702. doi: 10.7554/eLife.34702. PMID: 29714687; PMCID: PMC5976440.

Response 2: Thank you for your comment. Although jasmonic acid signal is highly similar to auxin signal (Han, 2017), JA signal has more advantages as an inducer. As an important regulatory signal of plant physiological processes, auxin signaling is widely involved in the growth and development of plants and the synthesis and regulation of primary metabolites (Westfall et al., 2013). While JA can inhibit plant growth, the inductive regulatory properties of JA signaling are more inclined to respond to stress responses and the synthesis of secondary metabolites (Westfall et al., 2013). Both JA and auxin are bound to amino acids in both lower and higher land plants, JA-Ile is the biologically active form of JA, and amino acid-bound auxin may become inactive (Ludwig-Müller, 2011). On the other hand, the synthesis and regulation of many natural products (such as gossypol, paclitaxel, vinblastine) have a common regulatory pathway, the jasmonic acid pathway, and jasmonic acid can be expressed in Escherichia coli and yeast. When methyl jasmonate is applied externally, it can regulate the heterologous expression of microbial host, so jasmonate signal has better specificity than auxin signal. In addition, JA can be rapidly activated by mechanical damage, and its induction is much simpler than that of auxin. Moreover, the JA signal can also form volatiles that spread to nearby plants, doubling the effect. Therefore, heterologous reconstruction of JA regulation and the establishment of a jasmonic acid-inducible promoter are more conducive to the efficient induction and expression of medicinal products.

Comment 3: Please consider mentioning the effect of endogenous jasmonic acid produced by the host organism on regulation by MYC2. Several microalgae biosynthesize jasmonic acid and conserve jasmonic acid signaling factors (Han 2017). Jasmonic acid has also been detected in E. coli and yeast, albeit in older literature (Abdala et al., 1999).

  1. Abdala, O. Miersch, N. Correa & S. Rosas (1999) Detection of Jasmonic Acid in Cultures of Escherichia Coliand Saccharomyces Cerevisiae, Natural Product Letters, 14:1, 55-63, DOI: 10.1080/10575639908045435

 Guan-Zhu Han, Evolution of jasmonate biosynthesis and signaling mechanisms, Journal of Experimental Botany, Volume 68, Issue 6, 1 March 2017, Pages 1323–1331, https://doi.org/10.1093/jxb/erw470

Response 3: Thank you for your suggestion. We add “JA exists in seed plants, and JA and JA-Ile are also detected in bryophytes [131]. And JA was even detected in Escherichia Coli and Saccharomyces Cerevisiae cultures [132]. The JA-JAZ-MYC2 system activates the corresponding biosynthetic system by applying high doses of JA. Under non-stress conditions, the impact of the low content of endogenous JA produced in the host organism on the JA-JAZ-MYC2 system may be inevitable but also limited. The widespread existence of JA suggests that it may be feasible to use the JA-JAZ-MYC2 system to regulate biosynthesis in eukaryotic and prokaryotic hosts.” in the paragraph 1 of section 6 (L441-L447).

Comment 4: In Figures 2a-c, the scientific names are used, but in Figures 2d-e, the metabolite names are used. The names listed should be consistent.

Response 4: Thank you for your comment. We have modified Figure 2 and re-upload the modified image.

Reference

Abdala, G., Miersch, O., Correa, N., and Rosas, S. (1999). Detection of jasmonic acid in cultures of Escherichia coli and Saccharomyces cerevisiae. Nat. Prod. Lett. 14, 55–63. doi: 10.1080/10575639908045435.

Cui, Y., Mao, R., Chen, J., and Guo, Z. (2019). Regulation mechanism of MYC family transcription factors in jasmonic acid signalling pathway on taxol biosynthesis. Int. J. Mol. Sci. 20. doi: 10.3390/ijms20081843.

Dhanasekaran, R., Deutzmann, A., Mahauad-Fernandez, W. D., Hansen, A. S., Gouw, A. M., and Felsher, D. W. (2022). The MYC oncogene — the grand orchestrator of cancer growth and immune evasion. Nat. Rev. Clin. Oncol. 19, 23–36. doi: 10.1038/s41571-021-00549-2.

Du, M., Zhao, J., Tzeng, D. T. W., Liu, Y., Deng, L., Yang, T., et al. (2017). MYC2 orchestrates a hierarchical transcriptional cascade that regulates jasmonate-mediated plant immunity in tomato. Plant Cell 29, 1883–1906. doi: 10.1105/tpc.16.00953.

Fu, G., Yue, J., Li, D., Li, Y., Lee, S. Y., and Zhang, D. (2022). An operator-based expression toolkit for Bacillus subtilis enables fine-tuning of gene expression and biosynthetic pathway regulation. Proc. Natl. Acad. Sci. U. S. A. 119, 1–11. doi: 10.1073/pnas.2119980119.

Gossen, M., and Bujardt, H. (1992). Tight Control of Gene Expression in Mammalian Cells by Tetracycline-Responsive Promoters. Proc. Natl. Acad. Sci. 89, 5547–5551. doi: 10.1073/pnas.89.12.5547.

Han, G. Z. (2017). Evolution of jasmonate biosynthesis and signalling mechanisms. J. Exp. Bot. 68, 1323–1331. doi: 10.1093/jxb/erw470.

Heiss, S., Hörmann, A., Tauer, C., Sonnleitner, M., Egger, E., Grabherr, R., et al. (2016). Evaluation of novel inducible promoter/repressor systems for recombinant protein expression in Lactobacillus plantarum. Microb. Cell Fact. 15, 1–18. doi: 10.1186/s12934-016-0448-0.

Holtz, W. J., and Keasling, J. D. (2010). Engineering Static and Dynamic Control of Synthetic Pathways. Cell 140, 19–23. doi: 10.1016/j.cell.2009.12.029.

Kazan, K., and Manners, J. M. (2013). MYC2: The master in action. Mol. Plant 6, 686–703. doi: 10.1093/mp/sss128.

Ludwig-Müller, J. (2011). Auxin conjugates: Their role for plant development and in the evolution of land plants. J. Exp. Bot. 62, 1757–1773. doi: 10.1093/jxb/erq412.

Niu, T., Liu, Y., Li, J., Koffas, M., Du, G., Alper, H. S., et al. (2018). Engineering a Glucosamine-6-phosphate Responsive glmS Ribozyme Switch Enables Dynamic Control of Metabolic Flux in Bacillus subtilis for Overproduction of N-Acetylglucosamine. ACS Synth. Biol. 7, 2423–2435. doi: 10.1021/acssynbio.8b00196.

Purcell, O., Peccoud, J., and Lu, T. K. (2014). Rule-based design of synthetic transcription factors in eukaryotes. ACS Synth. Biol. 3, 737–744. doi: 10.1021/sb400134k.

Scossa, F., Benina, M., Alseekh, S., Zhang, Y., and Fernie, A. R. (2018). The Integration of Metabolomics and Next-Generation Sequencing Data to Elucidate the Pathways of Natural Product Metabolism in Medicinal Plants. Planta Med. 84, 855–873. doi: 10.1055/a-0630-1899.

Shoji, T., and Yuan, L. (2021). ERF Gene Clusters: Working Together to Regulate Metabolism. Trends Plant Sci. 26, 23–32. doi: 10.1016/j.tplants.2020.07.015.

Urlinger, S; Udo, B; Marion, T; Mazahir,H. (2000). Exploring the Sequence Space for Tetracyline-Dependent Transcriptional Activators: Novel Mutations Yield Expanded range and Sensitivity. Proc. Natl. Acad. Sci. 14, 7963-68. doi:10.1073/pnas.130192197.

Verhage, A., Vlaardingerbroek, I., Raaymakers, C., Van Dam, N. M., Dicke, M., Van Wees, S. C. M., et al. (2011). Rewiring of the Jasmonate signaling pathway in arabidopsis during insect herbivory. Front. Plant Sci. 2. doi: 10.3389/fpls.2011.00047.

Westfall, C. S., Muehler, A. M., and Jez, J. M. (2013). Enzyme action in the regulation of plant hormone responses. J. Biol. Chem. 288, 19304–19311. doi: 10.1074/jbc.R113.475160.

Zheng, X. Y., Spivey, N. W., Zeng, W., Liu, P. P., Fu, Z. Q., Klessig, D. F., et al. (2012). Coronatine promotes pseudomonas syringae virulence in plants by activating a signaling cascade that inhibits salicylic acid accumulation. Cell Host Microbe 11, 587–596. doi: 10.1016/j.chom.2012.04.014.

Zhou, L. B., and Zeng, A. P. (2015). Engineering a Lysine-ON Riboswitch for Metabolic Control of Lysine Production in Corynebacterium glutamicum. ACS Synth. Biol. 4, 1335–1340. doi: 10.1021/acssynbio.5b00075.

Round 3

Reviewer 4 Report

The revised manuscript has addressed the issues pointed out in a previous review. The reviewer determines that the manuscript is acceptable for publication.